# Projections of faster onset and slower decay of El Niño in the 21st century

Hosmay Lopez [1✉], Sang-Ki Lee [1], Dongmin Kim [1,2], Andrew T. Wittenberg [3] & Sang-Wook Yeh [4]

Future changes in the seasonal evolution of the El Niño—Southern Oscillation (ENSO) during its onset and decay phases have received little attention by the research community. This work investigates the projected changes in the spatio-temporal evolution of El Niño events in the 21$^{st}$ Century (21 C), using a multi-model ensemble of coupled general circulation models subjected to anthropogenic forcing. Here we show that El Niño is projected to (1) grow at a faster rate, (2) persist longer over the eastern and far eastern Pacific, and (3) have stronger and distinct remote impacts via teleconnections. These changes are attributable to significant changes in the tropical Pacific mean state, dominant ENSO feedback processes, and an increase in stochastic westerly wind burst forcing in the western equatorial Pacific, and may lead to more significant and persistent global impacts of El Niño in the future.

[1] Atlantic Oceanographic and Meteorological Laboratory, NOAA, Miami, FL, USA. [2] Cooperative Institute for Marine and Atmospheric Studies, University of Miami, Miami, FL, USA. [3] Geophysical Fluid Dynamics Laboratory, NOAA, Princeton, NJ, USA. [4] Department of Marine Science and Convergent Technology, Hanyang University, Ansan, Korea. ✉email: Hosmay.lopez@noaa.gov

El Niño - Southern Oscillation (ENSO) is the dominant mode of interannual ocean-atmospheric variability in the tropical Pacific. Through its atmospheric teleconnections, ENSO is also the major source of seasonal predictability of global climate and extreme events[1–4]. Previous studies based on the climate models participating in the Coupled Model Intercomparison Project Phase 5 (CMIP5) have suggested an increase in the frequency of extreme El Niño events in the 21st Century (21 C) in response to increasing greenhouse gases[5–8], as well as an increase in ENSO amplitude[9], and associated rainfall variability[10–12], and further validated based on CMIP6[13]. Several studies have attributed these shifts in El Niño frequency and amplitude to the projected changes in the tropical Pacific mean state. Specifically, the majority of CMIP6 models show a projected mean state that mimics an El Niño-like condition (Supplementary Fig. S1), with a weaker zonal SST gradient and warming in the tropical Pacific, as a response to anthropogenic climate change (ACC). There have been several theories outlined to explain such mean state changes. One theory involves a weakening of the hydrological cycle in response to ACC would slow the Walker circulation, warming eastern Pacific SSTs[14–16]. Other theories call for a stronger evaporative damping of SST changes in the warm pool than in the cold tongue[17,18], or different cloud radiative feedbacks in the western and eastern tropical Pacific[19–21]. On the contrary, other studies have called for La Niña-like mean state changes under ACC, owing to oceanic feedbacks—such as the ocean thermostat mechanism, in which enhanced future thermal stratification acts through equatorial upwelling to enhance the zonal SST gradient, thus leading to a La Niña-like mean state change[22].

A future increase in equatorial Pacific upper-ocean thermal stratification and the associated surface warming of the eastern equatorial Pacific could increase the sensitivity of SSTs to fluctuations in local winds, upwelling, and thermocline depth, thereby enhancing the atmosphere-ocean coupling and the Bjerknes feedback loops[23] and boosting the amplitude of El Niño in the eastern equatorial Pacific[9]. This El Niño-like pattern of equatorial Pacific SST warming would reduce the near-equatorial zonal and meridional sea surface temperature (SST) gradients, facilitating shifts of west Pacific convection toward the eastern equatorial Pacific, and potentially increasing the likelihood of extreme El Niño events[5,24]. In addition, a recent study found that a future La Niña-like pattern of SST change would increase the zonal temperature gradient and zonal advective feedback in the central equatorial Pacific, potentially increasing the frequency and amplitude of strong El Niño events[7]. That is, *either* El Niño-like *or* La Niña-like mean state changes could increase the likelihood of extreme El Niño events. While most of these projected changes in the tropical Pacific climatology and variability have yet to emerge clearly in instrumental observations[25,26], paleo proxy evidence does suggest that ENSO has significantly strengthened in recent decades relative to prior centuries[27,28].

Previous studies have focused mainly on future changes of ENSO during the peak phase in boreal winter, or on changes in the diverse spatial patterns of ENSO SST anomalies[29,30]. Less attention has been paid to potential changes in the seasonal evolution of ENSO during the onset and decay phases with a few exceptions[8,31]. The onset and decay phases of ENSO, which typically occur in boreal spring and summer, have direct impacts on global climate variability and extreme events, including the Southeast Asian monsoon[32], tropical cyclones[33,34], and tornado outbreaks in the United States[35]. Therefore, there is an urgent need to better understand not only the spatial changes but also the temporal changes in ENSO characteristics arising from increasing greenhouse gases (GHGs), and the impacts of those ENSO changes on climate variability. This study examines a large suite of climate simulations—including the Community Earth System Model Large Ensemble (CESM-LENS), and available simulations from the Climate Model Intercomparison Project phase 6 (CMIP6), both driven by historical and projected future radiative forcing—in order to better understand potential future changes in El Niño's onset, decay, and remote impacts. Details on the models and their representation of ENSO are provided in the Methods section and Supplementary Material.

## Results

**Faster growth and slower decay of El Niño events in the 21st Century**. Similar to most of the CMIP5 models, the CESM-LENS and CMIP6 response to increasing GHGs is an El Niño-like warming pattern in the tropical Pacific[36] (Supplementary Fig. 1). The CESM-LENS and CMIP6 also project an increase in El Niño amplitude and likelihood of occurrence (Supplementary Table S2 and S3). In total, we found 350 El Niño events for the late 20th century (i.e., 1951–2000 or 20 C hereafter), and 419 events for the late 21st century (i.e., 2051–2100 or 21 C hereafter) in the 30 ensemble members from CESM-LENS. This indicates a 20% increase in the occurrence of El Niño. Similarly, we found 199 El Niño events in the 16 CMIP6 models for the 20 C period, and 234 events for the 21 C period. This indicates an increase of 17% in El Niño likelihood. Coincidentally, this increase in the occurrence of El Niño in CESM-LENS and CMIP6 models (i.e., 17–20%) is similar to the reported increase in occurrence of extreme El Niño events based on the results derived from CMIP5 models[5]. This increase is significant when compared to the expected natural variability (Supplementary Fig. 2), and suggests that anthropogenic influence on El Niño occurrence will emerge from natural variability by the middle of the 21 C under the most pessimistic scenario (i.e., RCP8.5 and/or Shared Socioeconomic Pathways; SSP585 scenario).

The projected changes in El Niño SSTAs between 20 C and 21 C (Fig. 1c, d) clearly show that El Niño develops much faster east of the dateline. Warm El Niño SSTAs in the far eastern Pacific persist throughout boreal spring in 21 C, extending the El Niño peak by about two to three months relative to 20 C, consistent with Carréric et al.[8] based on CESM-LENS and now reported here in CMIP6. However, warm El Niño SSTAs in the central Pacific (west of the date line) dissipate more rapidly during boreal winter and spring. These projected changes mark a contrast in El Niño SSTA evolution and a projected enhancement and eastward shift in convection over the tropical Pacific (Fig. 1g, f). In addition, the difference in SSTA evolution from 20 to 21 C suggests an increased tendency for eastward propagation of the SSTA under ACC, consistent with previous work, owing to a weakening of the Walker Circulation and associated eastward shift in convection[37,38]. It also suggests an increase in El Niño to La Niña transition under greenhouse warming (Fig. 1), consistent with previous work[38,39]. There is also a tendency for future El Niño events to evolve more strongly in the eastern Pacific as shown in Fig. 1c for CMIP6 and Fig. 1d for CESM-LENS when comparing the differences in spatio-temporal SSTA evolution between 21 C and 20 C. This is further supported by the eastward shift in convection and precipitation signals shown in Fig. 1g (CMIP6) and Fig. 1h (CESM-LENS).

The projected spatio-temporal changes in El Niño are summarized in Supplementary Tables S2 and S3 for CESM-LENS and CMIP6 respectively. Here the onset, duration, and demise months are quantified based on SSTA exceeding 0.5 °C for different ENSO indices[40]. There is a consistent increase in El Niño amplitude for all three major ENSO indices (e.g., Niño3, Niño3.4, and Niño4). While the start date of the events, appears to be model dependent, there is a consistent shift in the peak month from January to December based on Niño3.4 and Niño4. This, along

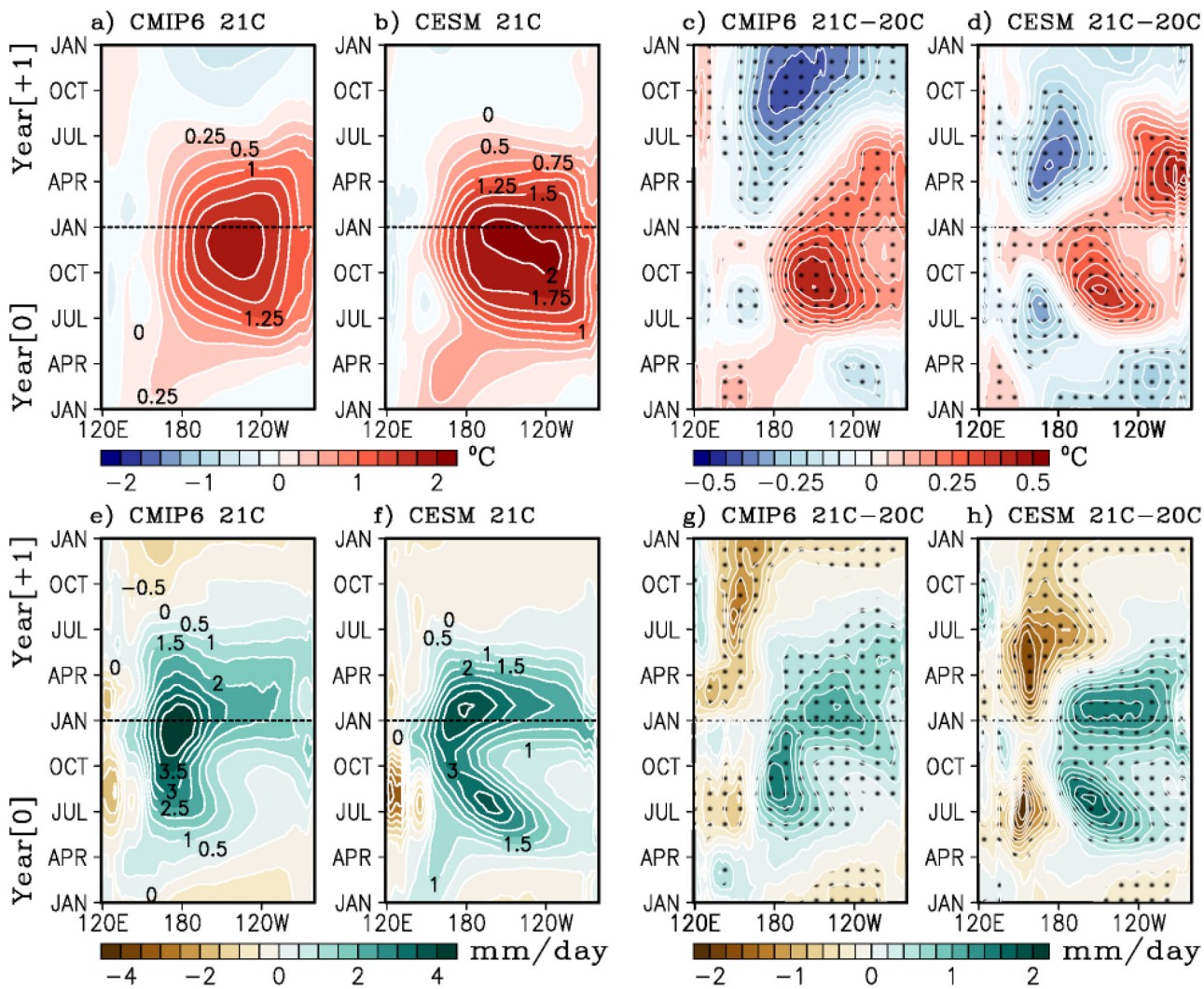

**Fig. 1 Spatio-temporal evolution of El Niño events.** Longitude-time diagrams of monthly sea surface temperature anomalies [top row, °C] in the equatorial Pacific (averaged 5°S–5°N), composited over El Niño events from **a** CMIP6 21st Century (21 C) ensemble mean (2051–2100) and **b** CESM-LENS 21 C (2051–2100). Also shown are the 21 C minus 20th Century (20 C, 1951–2000) difference from **c** CMIP6 and **d** CESM-LENS. Similarly, panels **e**–**h** show the composites of precipitation anomalies [mm day$^{-1}$]. Ordinate represents time from January of the onset year (Year 0) to December of the decay year (Year 1). Hatching on panels **c**, **d**, **g**, and **h** indicates statistical significance at the 95% confidence level using a bootstrapping technique (see Methods).

with the projected higher amplitude of future events explains the faster growth rate of 21 C El Niño events compared to 20 C events. There is a projected decrease in the decay rate over the Niño3 region, with later end day of the events. In contrast, the decay rate over the Niño4 region is projected to increase, with earlier termination of the events. The projected persistence of the future El Niño events into the boreal spring over the eastern Pacific could have marked repercussions on remote teleconnections, as it has been shown that SSTA over the eastern Pacific would enhance air-sea coupling, an eastward shift in atmospheric convection, and a potential larger remote influence[41]. While the projected increase in the number of El Niño events reported here occur in all types of El Niño, those events that persist into the boreal spring are projected to increase by 47%, dominating the reported increase of El Niño occurrence (Supplementary Fig. 3).

**What drives the projected changes in El Niño during the developing and decay phases?**
*Mixed layer heat budget analysis.* The seasonal El Niño evolution is analyzed using a mixed layer heat budget for the CESM-LENS (we left out the heat budget analysis for the CMIP6 for future

work). Figure 2 depicts the composite evolutions of heat storage, three feedback terms (i.e., thermocline, zonal advective, and Ekman feedbacks), and a residual term (see Methods). Note that the largest differences between 20 C and 21 C are in the amplitude rather than spatiotemporal structure. Specifically, the thermocline, zonal advective, and Ekman feedbacks are greatly enhanced (points labeled A in Fig. 2) in the eastern equatorial Pacific during the developing phase in boreal summer and fall (Year 0), readily explaining the faster rise in heat content (Fig. 2 green contour) and the faster growth of SSTAs during the growth phase (Fig. 1c). During January-March (Year +1), the Ekman feedback is enhanced in the eastern equatorial Pacific between 120°W and 80°W (point labeled B in Fig. 2). A similar projected increase in the Ekman feedback is also found during the decay phase in boreal winter, which is in contrast with a negative Ekman feedback in the 20 C. Interestingly, the thermocline feedback is reduced in the central equatorial Pacific between 170°E and 140°W during the decay phase in boreal spring (point labeled C in Fig. 2). Overall, the residual term, which is mostly dominated by the net surface heat fluxes (Supplementary Fig. 4), generally constitutes a damping of the heat content anomaly.

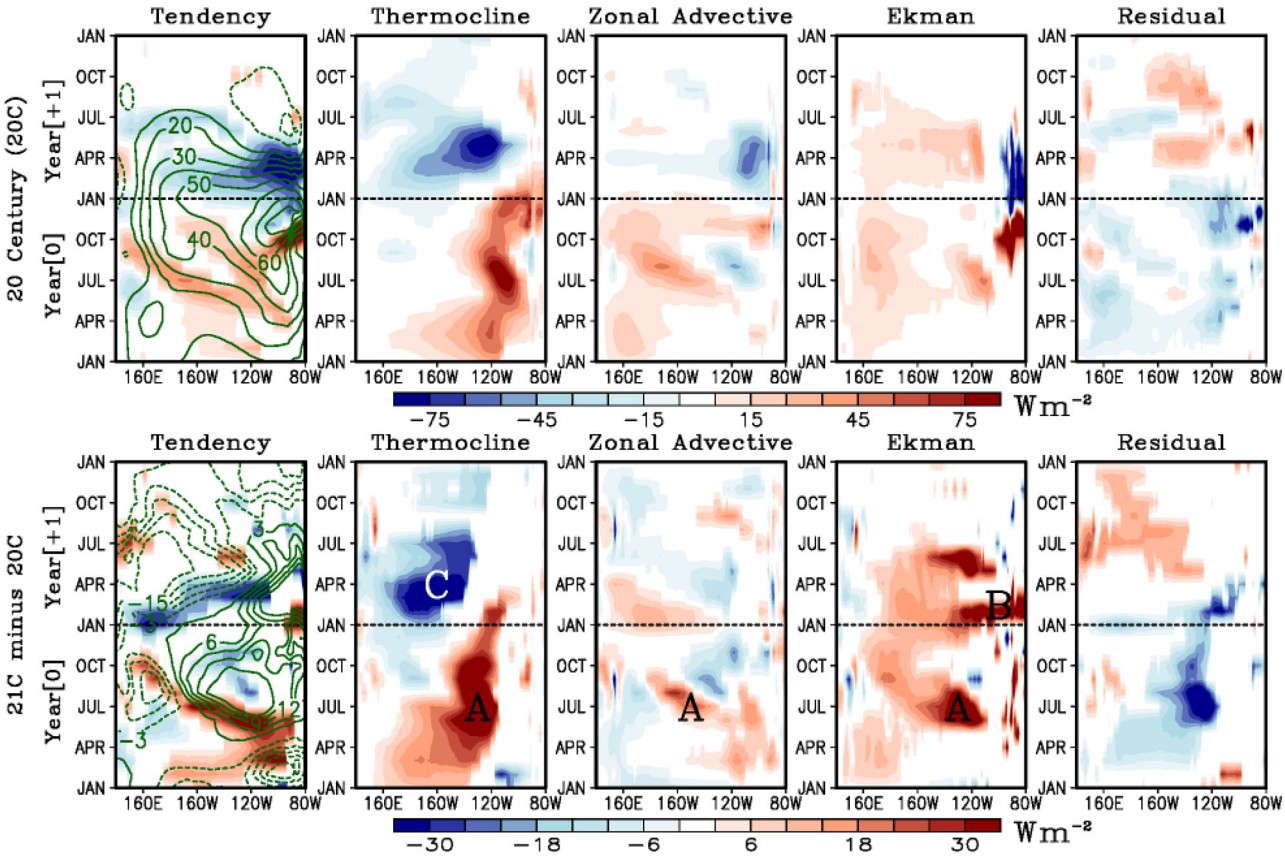

**Fig. 2 Mixed layer heat budget analysis during El Niño events for CESM-LENS.** The top row shows the composites for the heating tendency, thermocline feedback, zonal advective feedback, Ekman feedback, and residual term from Eq. 1 (color, W m$^{-2}$) and the heat content anomaly (green contours, 10$^8$ J m$^{-2}$) for the 20$^{th}$ Century (20 C). Bottom row shows the projected future change in the composite, for the 21$^{st}$ minus the 20$^{th}$ Century (21 C). All terms are computed assuming a constant 75 m mixed layer depth. See text for references to points A, B, and C. Only showing composite anomalies exceeding the 95$^{th}$ percentile significance level based on a bootstrapping technique (see Methods).

Nonlinear processes have been shown to be important for understanding the ENSO response to mean state changes from anthropogenic forcing[9] and also for their role in modulating the mean state changes[9,13], as they govern characteristics of extreme ENSO events[13,37,42]. Therefore, the residual term in the heat budget is further decomposed into three nonlinear temperature advection terms (Supplementary Fig. 4). Overall, the contribution of the nonlinear terms to the heat budget analysis is much smaller in amplitude than the linear ENSO feedback terms (e.g., thermocline, zonal advective, and Ekman feedback terms, Fig. 2), when averaged across all El Niño events in general. However, a projected increase in the nonlinear meridional advection may support the faster growth of future El Niño events, whereas an increase in the nonlinear zonal advection may support the persistence of future El Niño events into the boreal spring in the eastern Pacific.

To further investigate what dynamical processes are responsible for the projected changes in feedbacks and thus El Niño growth and decay, we repeat the computations of the feedback terms in Eq. 1 (Methods) by either fixing the mean state (i.e., overbar) or the anomalies (i.e., primes) to that of the 20$^{th}$ Century (Supplementary Table 4 and Supplementary Figs. 5 and 6). The projected changes in the thermocline feedback are mostly explained by an increase in the vertical gradient of anomalous temperature ($\frac{\partial T'}{\partial z}$ in Supplementary Figs. 5 and 6d) rather than the projected reduction in mean upwelling ($\bar{w}$ in Supplementary Figs. 5 and 6g). In addition, the zonal advective feedback changes are dominated by an increase in the anomalous zonal current ($u'$, Supplementary Figs. 5 and 6e)

rather than changes in the mean zonal temperature gradient ($\frac{\partial \bar{T}}{\partial x}$, Supplementary Figs. 5 and 6h). Whereas, the projected increase of the Ekman feedback can be explained by both contributions from anomalous upwelling ($w'$, Supplementary Figs. 5 and 6f) and an increase in the mean stratification ($\frac{\partial \bar{T}}{\partial z}$, Supplementary Figs. 5 and 6i). While $w'$ is mostly in the central Pacific, $\frac{\partial \bar{T}}{\partial z}$ dominates in the eastern Pacific. As can be seen from Supplementary Fig. 6f, changes in $w'$ lead to enhanced Ekman feedback in the eastern Pacific, because this is where $w'\frac{\partial \bar{T}}{\partial z}$ is strongest. The contribution from projected changes in $\frac{\partial \bar{T}}{\partial z}$ as shown in Supplementary Fig. 6 is mostly during the growth phase of El Niño, supporting faster growth of the events. In contrast, changes in the anomalous upwelling $w'$ are more relevant to the persistence of the events into the boreal spring (i.e., decay phase).

Analysis of future projections of ENSO SSTAs in the CMIP6 models has shown an increase in eastward SSTA propagation during the decay phase of El Niño[13]. It is well known that zonal SSTA propagation during the termination of El Niño is determined by the competing influences between the thermocline feedback, zonal advective and Ekman feedbacks[43]. Specifically, the thermocline feedback tends to drag SSTAs eastward, while the zonal advective and Ekman feedbacks tend to drag SSTAs westward, as evident in Figs. 1 and 2. Therefore, the projected increase in positive thermocline feedback east of 140°W and negative thermocline feedback west of 140°W (box C in Fig. 2) along with the increased positive Ekman feedback between 120°W and 80°W during JAN-MAR(+1) (box B in Fig. 2)

explain why El Niño SSTAs persist longer in the eastern Pacific in the late 21 C, in contrast to a westward propagation of SSTAs typically observed during the late 20 C (Fig. 1b, f).

The stronger El Niño events during 21 C are associated with stronger westerly wind anomalies and anomalous eastward currents (u', Supplementary Fig. 5b), which in turn induce stronger anomalous downwelling. This process is usually maximized during the peak El Niño phase i.e., DEC-FEB (+1), as seen in Supplementary Fig. 5. In addition, as shown in a previous study[44], the anomalous meridional winds may also drive convergent surface currents that induce anomalous equatorial downwelling in the equatorial Pacific[45].

It is important to investigate what mechanism could explain the tendency to have a shift towards stronger eastern Pacific El Niño events in the late 21 C (Fig. 1). The thermocline feedback is one of the major contributors of strong El Niño events[46]. For example, the negative thermocline feedback is enhanced in the central Pacific west of 140°W during the decay phase (Fig. 2). Analysis of the projected changes in each component of the thermocline feedback suggests that while the mean upwelling is weakened due to weaker trade winds, $\frac{\partial T'}{\partial z}$ is projected to increase (Supplementary Figs. 5 and 6). These changes are potentially due to either (1) larger thermocline depth anomalies associated with stronger westerly wind anomalies in the future (Supplementary Fig. 7) and/or (2) a stronger dependence of $\frac{\partial T'}{\partial z}$ on thermocline depth anomalies. The mean thermocline is projected to deepens (shoals) in the eastern (western) Pacific and sharpen (e.g., increased $\frac{\partial \bar{T}}{\partial z}$) during the 21 C (Supplementary Fig. 8), strengthening the thermocline feedback.

The reduced mean upwelling is consistent with the projected reduction of the mean easterlies in the 21 C and the reduction in the zonal gradient of the mean thermocline (Supplementary Fig. 8a). These changes oppose the projected strengthening of the thermocline feedback and zonal advective feedback, which are instead dominated by the 21 C increase in the anomalous temperature gradient ($\frac{\partial T'}{\partial z}$; see Supplementary Figs. 5 and 6d) and background thermal stratification in the eastern Pacific (e.g., increased $\frac{\partial \bar{T}}{\partial z}$; see Supplementary Fig. 8). The enhanced thermocline and Ekman feedbacks amplify El Niño events towards the eastern Pacific during the decay phase (FEB + 1 to APR + 1, Fig. 1), where the thermocline feedback and Ekman feedback produce a positive tendency in the heat content, extending the event into the boreal spring in the eastern Pacific. The stronger westerly anomalies indicate stronger coupled feedbacks (i.e., Bjerknes feedback). That is, the stronger future El Niño SSTA anomalies induce more vigorous atmospheric convection, which in turn produce stronger equatorial westerly wind anomalies in the western and central Pacific (Supplementary Fig. 7f). These wind anomalies reinforce the thermocline feedback through equatorial downwelling Kelvin waves and the zonal advective feedback through mean temperature advections by anomalous zonal currents (Fig. 2), enhancing El Niño. This is consistent with previous work outlining a future increase in atmosphere-ocean coupling[9], and more effective thermocline feedback in the eastern equatorial Pacific[8], as also shown in Supplementary Fig. 7 here.

In summary, the projected changes in major El Niño feedbacks (e.g., increased positive thermocline feedback east of 140°W and negative thermocline feedback west of 140°W, and increased positive Ekman feedback between 120°W and 80°W) cause El Niño SSTAs to persist longer in the eastern Pacific in the late 21 C, in contrast to a westward propagation of SSTAs typically observed during the late 20 C (Fig. 1c, d).

*Impacts of a changing tropical Pacific climatology.* The projected increase in equatorial Pacific time-mean SST is the strongest over the cold-tongue region in boreal fall and winter (Fig. 3), the

seasons when a climatologically shallow thermocline and strong cold tongue tend to intensify the vertical and zonal temperature gradients in the equatorial central Pacific. This seasonal intensification is weakened in 21 C, leading to weaker vertical and zonal temperature gradients during boreal fall and winter. Additionally, the time-mean thermocline flattens along the equator, shoaling the thermocline in the west and deepening it in the east (Fig. 3e). The thermocline also shows more intense thermal stratification (Supplementary Fig. 8), particularly during May-October in the central equatorial Pacific. The time-mean precipitation is also projected to increase over the central Pacific, especially during May-October (Fig. 3f).

These future changes in the time-mean precipitation and thermocline in the central Pacific (160°E–120°W) during boreal summer provide a more favorable background environment to support faster and stronger development of El Niño, by enhancing the thermocline feedback. In the far eastern equatorial Pacific (100°W–80°W) in boreal spring and early summer of year +1 (decay year), the future climatology shows a deeper thermocline and more precipitation (Fig. 3e, f). These changes suggest that the far eastern equatorial Pacific in boreal spring provides more favorable background environmental conditions for active air-sea interactions and convective activity in 21 C, as evidenced by the projected enhancement of convective precipitation during 21 C El Niño events (Fig. 1), thus supporting the persistent El Niño SSTAs in the far eastern Pacific during the decay phase in boreal spring.

Strong eastern Pacific events that persist into the boreal spring are strongly coupled to the ITCZ[8]. A recent study showed that the interaction between the ITCZ and El Niño in boreal spring is a key element in determining the decay of El Niño[47]. In this season, the eastern Pacific SST reaches its annual maximum (minimum) in the Southern (Northern) Hemisphere; thus, the ITCZ is climatologically further south in boreal spring (Supplementary Fig. 9). This period also coincides with the decaying phase of El Niño in the eastern Pacific (Fig. 1). It is also important to note that during the boreal spring, atmospheric convection in the eastern Pacific is very sensitive to small changes in SST because it is near the convection threshold. Future projections from CESM-LENS show enhanced climatological precipitation and westerly wind anomalies along the ITCZ (21 C minus 20 C) especially during February-April (Supplementary Fig. 9c). In addition, deep convection associated with El Niño events in the 21 C is greatly enhanced and produces stronger westerly wind anomalies, allowing for more prolonged and eastward-intruding SST anomalies, weakening the mean upwelling and thus extending the persistence of El Niño events into boreal spring[48]. The stronger El Niño events in 21 C are also associated with stronger anomalous eastward currents (u', Supplementary Fig. 5b), which induce stronger anomalous downwelling. Therefore, the reduction in the equatorial Trade winds in the 21 C relative to 20 C (i.e., westerly anomalies) suppresses the equatorial upwelling. This, together with the eastern Pacific warming and enhanced stratification, slow down the decay of El Niño events in the 21 C in the eastern Pacific.

*Changes in Westerly Wind Bursts.* Previous studies have stressed the role of stochastic forcing in modulating ENSO growth, variability, and predictability. For instance, the stochastic optimal forcing pattern, the noise forcing pattern prone to lead to ENSO growth, is consistent with the spatial structure associated with observed Westerly Wind Bursts (WWBs)[49]. WWBs excite downwelling Kelvin waves that propagate eastward along the equator[50] and help generate warm SSTAs in the central and eastern equatorial Pacific during El Niño[51]. The characteristics of ENSO events are also strongly affected by WWBs[52–54]. For instance, WWBs more strongly modulate the thermocline

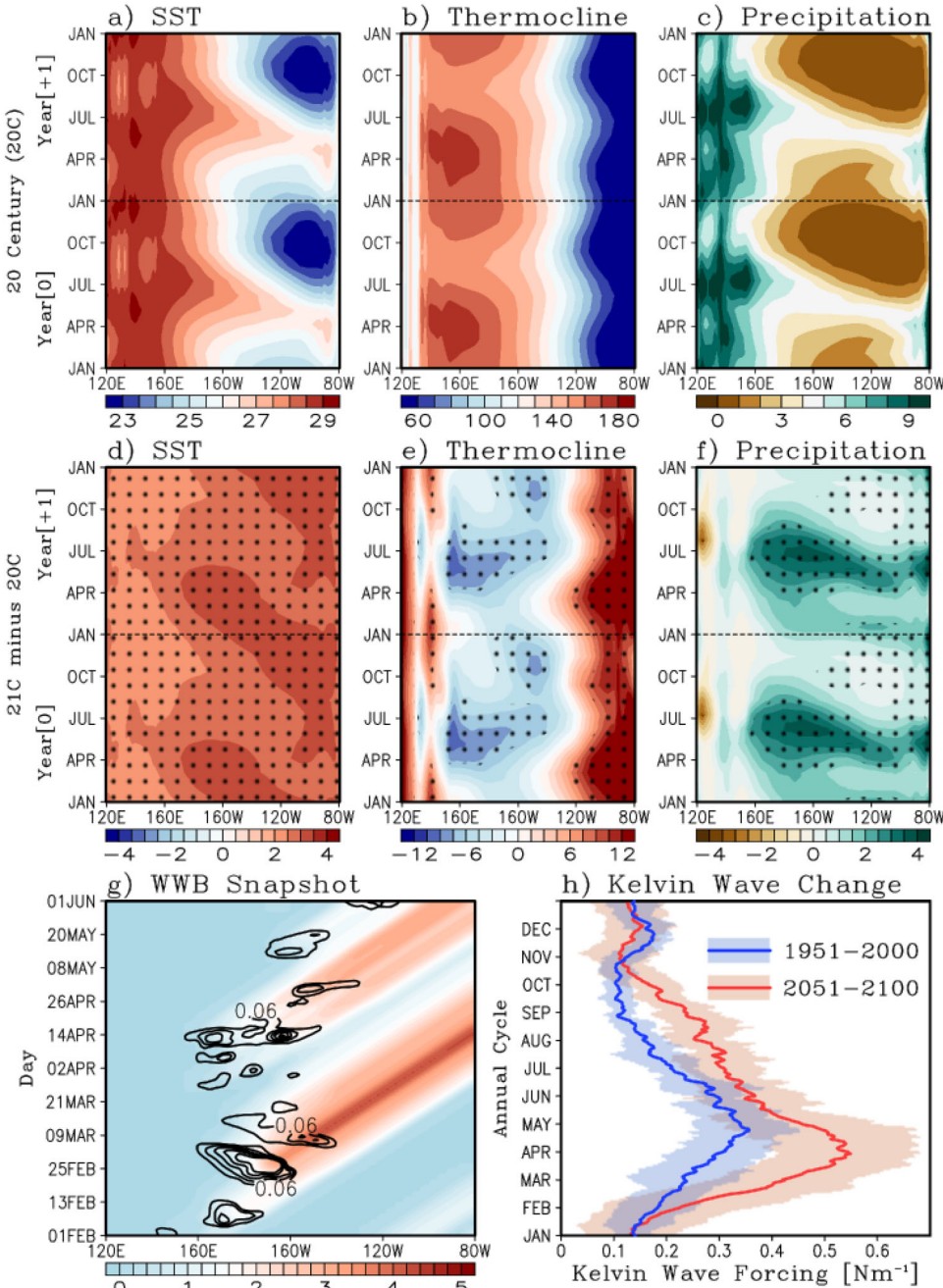

**Fig. 3 Equatorial Pacific climatology and projected changes, as simulated by CESM-LENS.** Two-year repeated monthly climatology of **a** sea surface temperature [°C], **b** thermocline depth [m], and **c** precipitation [mm/day], averaged 5°S-5°N during 1951–2000. Panels **d**-**f** show the projected change in climatology, 2051–2100 minus 1951–2000. The monthly climatology is computed as a time mean (for each calendar month) of the ensemble mean of all 30 ensemble members. Hatching indicates where the differences are statistically significant at the 95th percentile, based on a bootstrapping technique (see Methods). **g** A typical example illustrating the relationship between WWBs [contours, N m$^{-2}$] and the resulting downwelling equatorial Kelvin wave forcing [color shading, N m$^{-1}$] as estimated using Eq. 2. **h** Daily climatology of the downwelling equatorial Kelvin wave forcing for the 20th (blue) and 21st (red) Century integrated across the Pacific (Eq. 2). The thick lines represent the ensemble mean, while the shading corresponds to the ensemble spread (full min-max intra-ensemble range).

feedback than the zonal advective feedback, with important implications for El Niño diversity[55]. WWBs have very diverse sources, originating from the MJO activity, cold surges from mid-latitude, and tropical cyclones[56] and contribute to the spring prediction barrier for ENSO forecasts[57]. Also, these events have a deterministic component produced by the SSTAs associated with ENSO. However, it is not clear from the current literature how WWBs characteristics will change in the future[58,59].

Here, we look at the projected changes in WWBs activity in CESM-LENS. The WWB forcing and induced equatorial Kelvin wave response were computed using daily zonal wind stress ($\tau_x$) from the 20 C and 21 C simulations of the CESM-LENS (see Methods and Fig. 3g, h). Note that there is a significant projected increase in downwelling Kelvin wave forcing for the 21 C compared to the 20 C (Fig. 3h), suggesting an increase in WWBs activity and thus optimal wind stress forcing for El Niño

growth[49]. There is also a significant shift of this forcing to occur earlier in the year (Fig. 3h), from late April (20 C) to late March (21 C). This result is also consistent with the projected faster onset of El Niño (Fig. 1) and also with the projected increase in the thermocline feedback during the onset phase (Fig. 2). Enhanced WWB activity during boreal spring and enhanced air-sea coupling in the far eastern Pacific explain the faster growth rate and slower decay of El Niño in the 21 C.

WWBs events could be decomposed into a state independent (i.e., stochastic) and a state-dependent (i.e., semi-stochastic) components[53–55,60], where the latter is dependent on the underlying SST state. Both of these components could be enhanced in the future climate. For example, an amplification and eastward expansion of the climatological warm pool due to an increase in the mean SST would shift atmospheric convection eastward into the central Pacific. The peak season of WWBs activity coincides with boreal spring, when the warm pool is farthest eastward. The state-dependent component of the WWBs strengthens in the future as well, due to the future El Niño SSTAs reaching larger amplitudes, and the enhanced coupling between the ocean and atmosphere[9].

Previous studies have shown that the ITCZ position strongly influences the nonlinearity of the equatorial zonal wind response to SSTAs, which derives largely from WWBs state dependence[61,62]. Such nonlinearity, in turn, modifies ENSO phase asymmetries of event amplitude, duration, and transition[63,64], which are strongly linked to the El Niño spatiotemporal evolution discussed here. While most CMIP5 models do not show a discernible increase in state-dependent component of WWBs under anthropogenic forcing[60], the CESM model is one of the few that show an increase in WWBs occurrence as noted in this paper[60]. The CESM is also the only CMIP5 model that readily simulated the correct state-dependent component of WWBs[60].

## Changes in the remote impacts of El Niño in the 21st century
*Introduction and prior work*. El Niño-related anomalous precipitation and upper-level divergence modulate the extratropical circulation via atmospheric westward Rossby wave propagation[65], influencing weather and climate over North America[66]. However, ENSO teleconnection is strongly modulated by internal atmospheric variability[67], as well as the spatial details of the ENSO SSTAs in the tropical Pacific[68].

Previous studies have shown using several CMIP5 models a future projected increase in ENSO-driven precipitation of around 20% in regions that are largely impacted by ENSO signals[69]. Changes in ENSO teleconnections under ACC are strongly linked to both the changes in oceanic and atmospheric mean state and in ENSO amplitude and spatial pattern[70]. The SST-rainfall relationship over the eastern Pacific is expected to strengthen under ACC driven by the exponential relationship between temperature and vapor pressure through the Clausius-Clapeyron relationship[71]. Therefore, even if the ENSO amplitude remains similar to that of the 20 C, the response of tropical and extratropical atmosphere to ENSO will be intensified[72]. While future projections suggest a systematic strengthening in ENSO remote teleconnections[73], there is still large regional intermodel differences due to internal variability and model errors[3].

Besides the reported large uncertainties in future changes in ENSO teleconnections, the spatiotemporal changes in El Niño SSTA reported in this work can modulate the seasonally-dependent remote teleconnections. We investigate how remote effects of El Niño events will change in the future – owing to changes in both the spatial and temporal patterns with faster growth, larger amplitude, and extended persistence over the boreal spring. As such, remote teleconnections are quantified separately for the developing phase (September-October-November, SON) and for the decay phase (March-April-May, MAM).

The El Niño-driven atmospheric circulation changes, as measured by the composite of 500 hPa geopotential height, is dominated by the Pacific South American pattern during the developing phase[74] (Supplementary Fig. 10a) and the Pacific North American pattern during the decay phase[65] (Supplementary Fig. 10c). These patterns are primarily driven by the upper tropospheric divergence flow associated with tropical convection driven by El Niño. The tropical Pacific atmospheric forcing is projected to increase and shift eastward in the 21C[13], for the developing (Supplementary Fig. 10b) and decay phase (Supplementary Fig. 10d), leading to an intensification of the circulation anomaly. Future changes in El Niño's teleconnections during the developing phase (SON in Year 0) are mostly over the Southern Hemisphere (Supplementary Fig. 10b). This is due to the enhanced SSTAs and precipitation over the tropical Pacific (Fig. 1), and the preference of extra-tropical teleconnection to the winter hemisphere[75]. This is also consistent with the notion that the Southern Hemisphere ENSO response leads the ENSO peak in the tropics[76]. During the decay phase (MAM in Year +1), El Niño's remote effects mimic those of the peak phase, with enhanced future teleconnections.

*Remote impacts – North America*. The projections show a northeastward shift and intensification in temperature and precipitation anomalies over North America (Fig. 4 for CESM-LENS and Supplementary Fig. 11 for CMIP6), consistent with previous work[70,77]. This northeastward shift in the teleconnection is driven by the projected eastward shift of the PNA teleconnection pattern under global warming[78], resulting from a systematic eastward migration of the tropical Pacific convection centers associated with both El Niño and La Niña in the future[79]. The enhanced cooling is a result of increased clouds and precipitation over the southern U.S, with changes in the teleconnections being generally consistent among climate models[73]. The projected enhancement of the 20 C teleconnection into the 21 C is consistent between CESM-LENS and CMIP6, which showed that sufficiently warm and persistent SSTA in the far eastern equatorial Pacific is required to excite teleconnection patterns that influence rainfall over the western U.S[41].

*Remote impacts – South America*. ENSO influences South American climate through modulations of the Walker circulation as well as extra-tropical teleconnections (e.g., Rossby wave trains), producing a north-south dipole pattern in surface air temperature and rainfall with cooler and wetter (warmer and drier) conditions in the Southern (Northern) portion of the continent[80,81]. In addition, remote ENSO effects over South America strongly depend not only on amplitude but also the longitudinal location of maximum ENSO SSTA, with eastern Pacific events exhibiting a more pronounced shift in the Walker Circulation[82] and stronger extratropical teleconnections[74] than central Pacific events[81]. Consistent with the increased ENSO amplitude (especially in the eastern Pacific), ENSO's impacts over South America are also projected to increase in the 21 C, with a projected increase (reduction) in rainfall over Southeastern South America (Amazon basin)[81]. Over South America, El Niño's remote effects are also projected to increase, with enhanced warming over the northern two thirds of the continent and drier Amazon basin during both the developing (Fig. 4 for CESM-LENS and Supplementary Fig. 11 for CMIP6) and decay phases of El Niño (Fig. 5 for CESM-LENS and Supplementary Fig. 12 for CMIP6).

*Remote impacts – Australia*. El Niño is associated with warming and reduced precipitation over Australia during the historical

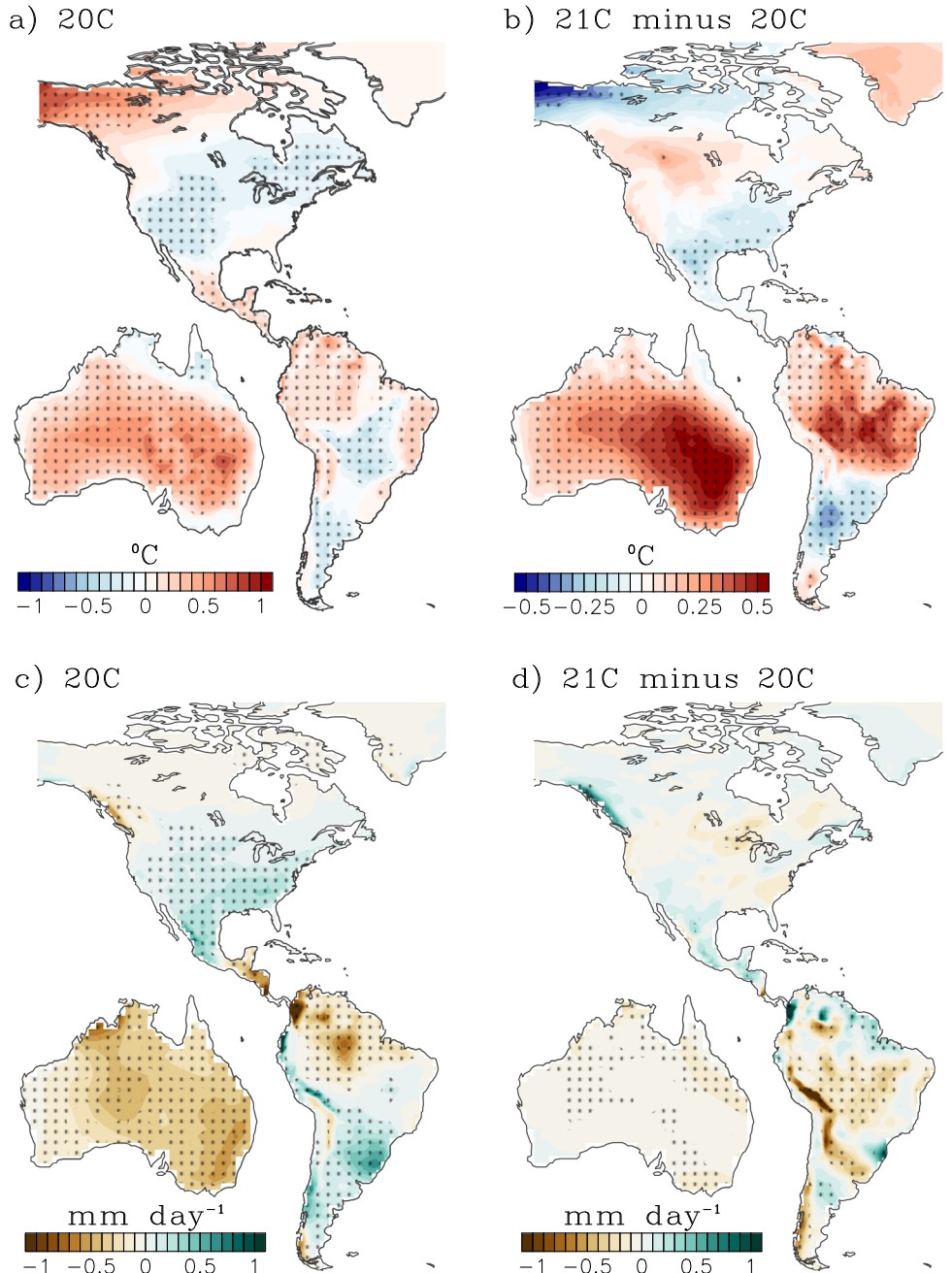

**Fig. 4 Representative map depicting the remote effects of El Niño event on surface temperature and precipitation and their projected changes from CESM-LENS. a** Surface temperature [blue-red, °C] and **c** precipitation [brown-green, mm day$^{-1}$] composite during September-October-November (SON, or developing year-0) for late 20$^{th}$ Century El Niño events (see Methods for Definition of El Niño). Projected changes in **b** temperature and **d** precipitation composite (21 C minus 20 C) during SON (or developing year-0). Stipples indicate anomalies that are significant at the 95% level based on a bootstrapping technique (see Methods).

period[73], especially over the north and eastern portions of Australia[83]. However, the ENSO response over Australia varies significantly with the location of maximum ENSO SSTA. For instance, central Pacific events tend to produce more widespread precipitation and temperature signals in Australia than eastern Pacific events[29]. There is a projected significant increase in El Niño-induced surface warming across the entire continent during SON (developing year-0; Fig. 4 for CESM-LENS and Supplementary Fig. 11 for CMIP6) and MAM (decay year+1; Fig. 5 for CESM-LENS and Supplementary Fig. 12 for CMIP6). Projected changes in rainfall are more pronounced during MAM, with more drying (wetting) of the northwestern (eastern) region,

consistent with the eastward shift in El Niño SSTA and precipitation anomalies in the tropical Pacific (Fig. 1).

## Discussion

The main objective of this work is to examine the dominant factors that alter the spatio-temporal evolution of El Niño events during the 21 C. Our major findings, based on CESM-LENS and CMIP6 model projections, are that El Niño in the late 21 C is projected to (1) grow at a faster rate, (2) persist longer over the eastern and far eastern Pacific, and (3) to produce larger and distinct remote impacts in surface temperatures and precipitation.

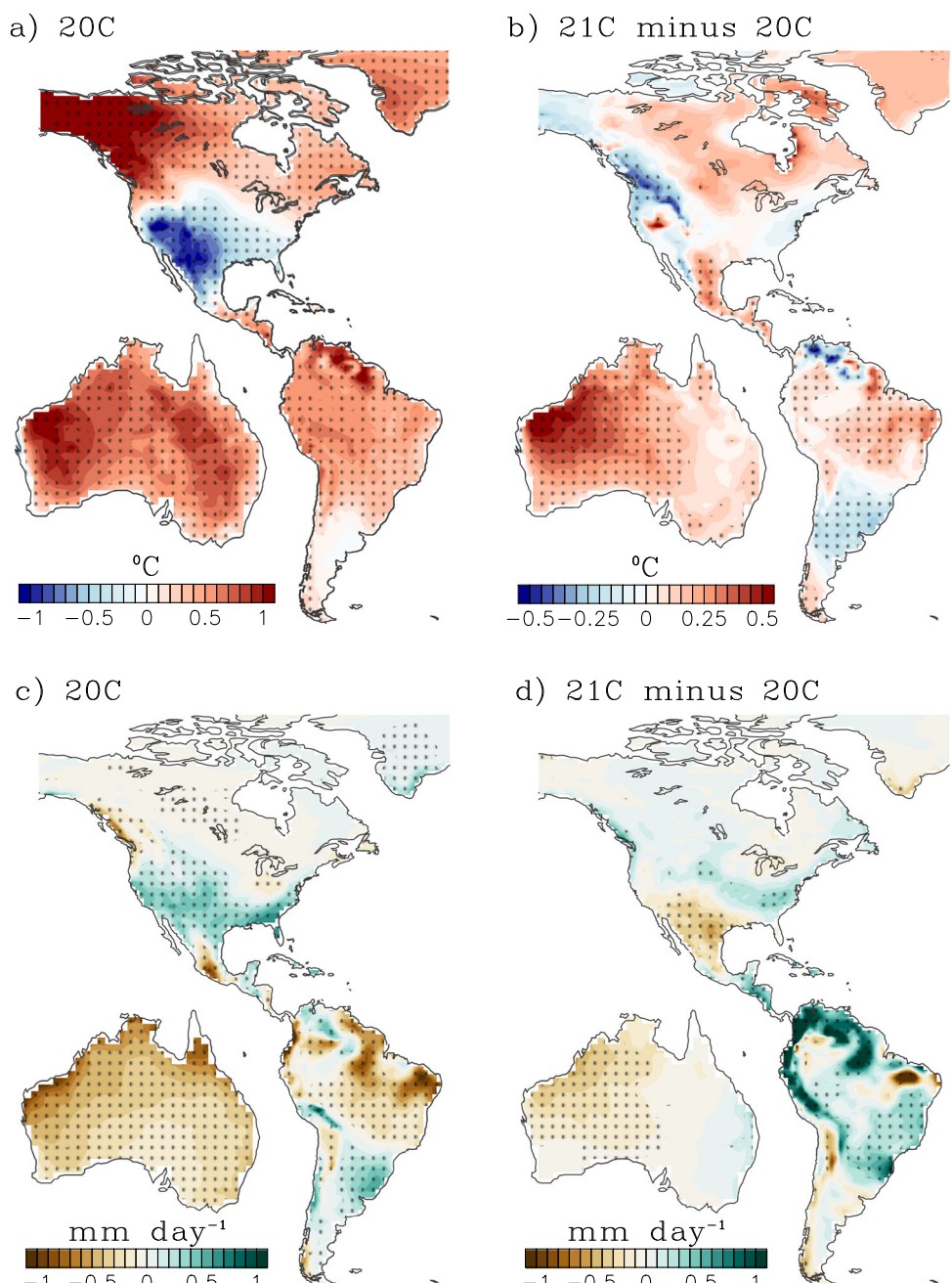

**Fig. 5 Representative map depicting the remote effects of El Niño event on surface temperature and precipitation and their projected changes from CESM-LENS. a** Surface temperature [blue-red, °C] and **c** precipitation [brown-green, mm day$^{-1}$] composite during March-April-May (MAM, or decay year +1) for late 20$^{th}$ Century El Niño events (see Methods for Definition of El Niño). Projected changes in **b** temperature and **d** precipitation composite (21 C minus 20 C) during MAM (or decay year+1). Stipples indicate anomalies that are significant at the 95% level based on a bootstrapping technique (see Methods).

These changes in El Niño are attributable to projected changes in the tropical Pacific climatology (including an increase in equatorial Pacific rainfall in boreal spring and summer, a shallower thermocline in the central Pacific, and a deeper thermocline in the far eastern Pacific in boreal spring), and associated changes in the dominant ENSO feedbacks (i.e., thermocline, zonal advective, and Ekman feedbacks). We also identified a significant projected increase in the stochastic forcing of El Niño from WWBs, and an earlier onset of the WWBs forcing in boreal spring. These changes in WWBs forcing are consistent with the projections of a faster onset for El Niño (Fig. 1). These changes in El Niño are consistent with a stronger equatorial upper-ocean recharge

process and a more effective thermocline feedback in the eastern Pacific[8], as well as an increased stratification in the eastern Pacific leading to stronger SSTA variance[9]. In addition, this work further suggests that the eastern Pacific persistence of future El Niño events is supported by enhanced Ekman feedback from stronger vertical stratification, and an increase in WWBs activity, a salient feature in the CESM-LENS.

Compared to the late 20 C, the likelihood of occurrence of El Niño in the late 21 C increases by about 20% in CESM-LENS and 17% in CMIP6, well outside the range expected from unforced internal variability. There is also a 47% increase in the number of El Niño events whose far eastern equatorial Pacific SSTAs persist

into boreal spring (Supplementary Fig. 3). Given that such long-lasting El Niño events have been linked to some of the largest climate impacts on North America[41], it is perhaps not surprising that we also find that the models project more significant and persistent extratropical teleconnections and remote climate impacts from El Niño in the future. A further implication of our results is that the lead time for skillful seasonal El Niño forecasts may be reduced in the future, due to the faster development of El Niño and a larger role for stochastic WWBs forcing. We are currently working to extend this analysis to future projections of La Niña.

We have presented results from the latest state-of-the-art climate model simulations of ENSO. Although these simulations have substantially improved over the past decade, a number of remaining model biases could affect the results presented here. Such biases include ITCZs displaced from their observed positions, an overly-intense cold tongue, unresolved stirring from tropical instability waves, and difficulties representing sub-gridscale processes and feedbacks (atmospheric convection and clouds, and near-surface ocean mixing), which may affect model projections of future ENSO behavior[84–93]. A recent study found that resolving mesoscale features in both the atmosphere and ocean produce a contrasting result when compared to low-resolution ENSO projections. That is, anthropogenic forcing induces a weakening of future ENSO variability[94]. It is noted that oceanic mesoscale features such as tropical instability waves, which are not resolved in low-resolution models, serve as an equally important damping mechanism for ENSO[90], as important as the thermodynamic and dynamic damping terms. However, there are still significant model biases even in eddy-resolving simulations[94], as well as important processes which are still parametrized. This, plus a need for longer simulations and multi-model ensembles, calls for further studies to confirm the contrasting ENSO projections between low- and high-resolution simulations.

Besides model biases, another limiting factor in studying future changes in ENSO is the difficulty in accurately separating the anthropogenic response of ENSO into its transient and equilibrium components. Analysis of millennial-scale warming in high emission scenarios from the Long Run Model Intercomparison have shown that the transient response to anthropogenic forcing contains large uncertainties, whereas the equilibrium response manifests a decrease in ENSO amplitude and no change in ENSO frequency[95]. In addition, the observed historical trend in Pacific mean state changes so far remains well within the unforced variability range[95]. Continued advances in these models will be crucial to achieve more reliable projections of future ENSO variability and its impacts on society.

## Methods

**Observations and model data availability**. For observations, we analyze the NOAA Extended Reconstructed Sea Surface Temperature version 5 (ERSST.v5)[96] and the rainfall from the Global Precipitation Climatology Project (GPCP version 2.3)[97]. The model simulations are taken from a 30-member ensemble simulation of the Community Earth System Model—Large Ensemble Simulation (CESM-LENS, data freely available at: https://www.cesm.ucar.edu/projects/community-projects/LENS/)[98]. The atmospheric component has 30 vertical levels with a horizontal resolution of 1.25° longitude by 0.94° latitude. The ocean model component has a 1° horizontal resolution with 60 vertical levels. CESM is one of the most skillful models in representing ENSO variability and associated global teleconnections[5,73]. We validated all relevant results using the CMIP6 model archive (data freely available at: https://esgf-node.llnl.gov/projects/cmip6/). All CMIP6 models were interpolated to the horizontal resolution of CESM-LENS. See Supplementary Table 1 for list of models.

The analysis comprises 16 CMIP6 models and 30 ensemble members from CESM-LENS under historical radiative forcing (i.e., during 1951–2000 or 20 C). The 21 C projection also comprises 30 ensemble members from CESM-LENS for the 2051–2100 period under the Representative Concentration Pathway 8.5 radiative forcing of the CMIP5 design protocol[99,100], and the Shared

Socioeconomic Pathways (SSP585) for the CMIP6 models. The RCP8.5 and SSP585 result in similar radiative forcing levels. Each model/ensemble member has a distinct climate trajectory due to differences in the atmospheric initial conditions. Differences among ensemble members from CESM-LENS are solely due to internal variability, while differences among models from CMIP6 also include model biases. We repeated relevant analyses with 10, 15, 20, and 30 members and the results converge beyond 15-members. Thus, adding more ensembles will not add much to the results qualitatively, beyond increasing the degrees of freedom[36].

All models employed here show reasonable representation of the spatio-temporal evolution of El Niño events (Supplementary Fig. 13), although there are some notable differences among individual models (Supplementary Fig. 14). The GFDL-CM4, GFDL-ESM4, HadGEM3, and CESM2 models show the most accurate representation of the spatio-temporal evolution of SSTA during El Niño events among the CMIP6 models. The ability of CESM-LENS to reproduce the observed evolution of El Niño is in general comparable to more up-to-date CMIP6 models.

Compared to observations, the CESM-LENS ensemble mean shows stronger and westward-displaced El Niño events, whereas the CMIP6 projections depict a weaker than observed El Niño amplitude (Supplementary Fig. 15). These model biases are common among many CMIP models[84,85]. Most of these differences in interannual variability can be traced back to biases in the mean state (Supplementary Figs. 1 and 2), such as the overly-intense and westward-displaced equatorial cold tongue, the double intertropical convergence zone (ITCZ), and the warm SST bias in the far eastern equatorial Pacific[101]. Simulating ENSO nonlinearity has also been deemed an important aspect when selecting models.

**Interannual variability definition in CESM-LENS and CMIP6**. For CESM-LENS, the ensemble mean is subtracted for each ensemble member, then a monthly climatology is further removed from the resultant anomalies. This is done to avoid the potential for each ensemble member having a different climatology. This method should objectively remove any trend (including non-linear) due to external forcing as well as CESM model bias. Thus, interannual anomalies for a given ensemble member exclude any trend associated with the increasing GHGs as well as responses of low-latitude volcanic eruptions[102,103], such as Pinatubo in 1991–92. For the CMIP6 models we used a slightly different approach, since unlike CESM-LENS, each CMIP6 model has its own physics and thus bias. To compute anomalies in CMIP6, we remove a 30-year running mean climatology derived for each CMIP6 models. Applying these two different approaches for removing the trend to CESM-LENS yielded nearly identical results (see Supplementary Fig. 16).

**El Niño event definition**. We define an El Niño event following the method used at the NOAA Climate Prediction Center where 3-month averaged SSTAs over the Niño3.4 region (e.g., area average from 170°W to 120°W and 5°S to 5°N) should exceed 0.5 °C for a minimum of five consecutive months. SSTAs are defined by removing the ensemble mean SST and monthly climatology for each ensemble member. The monthly climatology is defined for the period of 1951–2000 for the observations and historical model runs (20 C), and from 2051 to 2100 for the future projections (21 C).

**Mixed layer heat budget**. Equation 1 describes the mixed layer heat budget, where the mixed layer temperature tendency (term on the left-hand-side) is driven by the thermocline, zonal advective, and Ekman feedbacks as described by the right-hand-side integrands in Eq. 1 in that order, plus a residual term. Here, $T$ denotes the mixed layer temperature, and $u$ and $w$ are the zonal and vertical components of velocity, respectively, which are evaluated in an Eulerian reference frame on the model grid. $H = 75\ m$ is the mixed layer depth, taken here to be constant throughout the tropical Pacific[20]. We have tested these results using different mixed layer depths (i.e., *30, 50, 75,* and *100 m*) and found that the main conclusion is consistent and independent on the mixed layer depth. The entrainment velocity is taken to be the vertical velocity at the base of the mixed layer. Bars and primes represent the climatological mean and monthly anomalies, respectively, computed independently for the 20 C and 21 C. The residual term is composed of the surface net heat fluxes, meridional heat advection, non-linear advection (i.e., $\frac{\bar{v}\partial T'}{\partial x}$ ; $u'\frac{\partial T'}{\partial x}$ ; $w'\frac{\partial T'}{\partial z}$), diffusive heating, unresolved sub-grid scale heat fluxes, and sub-monthly scale heating.

$$\int_{-H}^{0} \frac{\partial T'}{\partial t} dz = -\int_{-H}^{0} \left[ \bar{w}\frac{\partial T'}{\partial z} + u'\frac{\partial \bar{T}}{\partial x} + w'\frac{\partial \bar{T}}{\partial z} \right] dz + Residual \qquad (1)$$

**Wind-forced equatorial oceanic Kelvin waves**. To identify WWBs, we first remove the 91-day running-mean climatology from the daily zonal wind stress. Then, the anomalous wind stress is averaged between 2.5°S and 2.5°N. Westerly wind stress anomalies exceeding +0.03 Nm$^{-2}$, with a minimum zonal fetch of 500 km, and a minimum duration of 3 days are defined as WWBs[54,55]. Since the influence of WWBs on equatorial dynamics depends on the amplitude, fetch, duration, and probability of occurrence of the forcing, a single parameter is defined here which encompasses all those aspects into one index and their integrated impact on ENSO. This index is summarized by Eq. 2 and is referred to as the downwelling Kelvin wave forcing, and thus is directly linked to El Niño

evolution[104]. Here, WWB($x,t$) represents the wind stress anomaly associated with WWBs, which is a function of longitude $x$, which is centered at $x_o$, and time $t$. Note that the Kelvin wave forcing is the integral of the WWB forcing from the central longitude $x_o$ to the eastern boundary $x_e$ along the characteristics (i.e., $\frac{x-x_0}{c}$) of equatorially trapped Kelvin waves with the observed $c = 2.4 \, ms^{-1}$ phase speed[104]. For the purpose of this work, variations in the parameter $c$ are not critical as we are not attempting to investigate the timing of forcing versus response, only the changes in the statistics of the forcing under anthropogenic influences.

$$\text{Kelvin Wave Forcing} = \int_{x_0}^{x_e} \text{WWBs}\left(x, \frac{x-x_0}{c}\right) dx \qquad (2)$$

**Statistical significance test – Bootstrapping technique**. A Monte Carlo bootstrapping method is employed to determine confidence intervals by subsampling the dataset. For example, all composite analyses presented are obtained by randomly selecting $r = 15$ out of $n = 30$ members from CESM-LENS with replacement (Eq. 3). This is done 1000 times in order to build a significant distribution of composites and assign 95th percentile confidence levels. The same approach is used when analyzing composites from CMIP6, by selecting $r = 8$ out of $n=16$ members. This yields over $10^8$ ($10^4$) possible combinations for CESM-LENS (CMIP6) respectively.

$$\binom{n}{r} = \frac{n!}{r!\,(n-r)!} = \text{possible combinations} \qquad (3)$$

## Data availability

Data related to this work can be downloaded from the websites listed below: CESM-LENS, data freely available at: https://www.cesm.ucar.edu/projects/community-projects/LENS/. CMIP6 model archive data freely available at: https://esgf-node.llnl.gov/projects/cmip6/. URLs for each individual model are provided in the Supplementary Information.

## Code availability

The GrADS, NCL, and Fortran codes used to perform the analyses can be accessed upon request to H.L.

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

## Acknowledgements

We would like to acknowledge Dr. Renellys Perez (NOAA/AOML) and Dr. Elizabeth Johns (NOAA/AOML) for their comments and suggestions that greatly improved the manuscript. This research was carried out in part under the auspices of the Cooperative Institute for Marine and Atmospheric Studies, a cooperative institute of the University of Miami and the National Oceanic and Atmospheric Administration (NOAA), cooperative agreement NA 20OAR4320472. Hosmay Lopez acknowledges support from NOAA/CPO/MAPP Award NA19OAR4310282.

## Author contributions

H.L. and S.K.L. conceived the study and H.L. wrote the initial draft of the paper. H.L., S.K.L., D.K., A.T.W., and S.W.Y. contributed to the design, the statistical analysis, and interpretation of the results as well as the writing of the final version of the paper.

## Competing interests

The authors declare no competing interests.
