## [Peer Review File · Nature Communications]

Projections of faster onset and slower decay of El Niño in the 21st centuryREVIEWER COMMENTS

Reviewer #1 (Remarks to the Author):

This is a nice study that investigates the response of El Niño to greenhouse forcing with a focus on the ENSO onset and decay phases. The issue is an important one to resolve given the significant impact of El Niño on climate, and the ways El Niño will change in a warming planet need to be understood. The study found that El Niño is projected to grow faster and persist longer over the eastern Pacific with significant remote impacts in air temperature and precipitation over the American continent and Australia. The analysis is quite thorough and comprehensive, but some clarifications are required and the writing can be improved, along with additional of relevant references for a more balanced paper.

A main critique would be a lack/omission of discussion on nonlinear ENSO processes which have been shown to be important for understanding the response to greenhouse forcing, e.g., see a review paper that recently came out: Cai et al. (2021) Nature Reviews Earth and Environment, <http://dx.doi.org/10.1038/s43017-021-00199-z>.

In this present analysis, the focus is on the linear terms, while the nonlinear terms are lumped together in a residual, along with air-sea heat fluxes. This may obscure their role in the different phases.

How well models simulate ENSO nonlinearity has also been deemed to be important. The inter-model consensus for increased ENSO SST variability has been based on models that reasonably simulate ENSO nonlinearity, and thus model selection is necessary. This was the case for CMIP5 models (Cai et al. 2018), but for CMIP6 it turns out that model selection is not quite necessary as ENSO nonlinearity is quite reasonable in these models (Cai et al. 2021). This is worth to be discussed.

There are studies that have concluded El Niño to La Niña transition being enhanced under greenhouse warming (e.g., An & Kim 2018, <https://agupubs.onlinelibrary.wiley.com/doi/full/10.1029/2018GL078476>). How to reconcile such result with the prolonged El Niño found in this present study?

Section 4 on El Niño impacts is rather short and rudimentary relative to the number of figures that they take up in the manuscript (50% of total figures). There's ought to be more discussions on how ENSO impacts on these regions with the relevant literature cited, including those that used CMIP5 models, e.g., Grose et al. (2020) for Australia <https://agupubs.onlinelibrary.wiley.com/doi/full/10.1029/2019EF001469>; Cai et al. 2020 on South America rainfall and references therein <https://www.nature.com/articles/s43017-020-0040-3>;

How the anomalies are obtained is not clear. For the CESM-LENS, it is understood that the ensemble mean can be subtracted from each member. But for the CMIP6 models, how is this done when every model has different climate sensitivity?

Please indicate statistical significance in Figure 2. In the caption, mention also it is longitude-time for the average over 5S-5N. There are other figures that do not have statistical significance prescribed (e.g., Fig. S8, S9).

L243-257: there seems to be no figure that explicitly shows changes in WWB activity, and no analysis that supports the statement on the enhancement of the stochastic and state-dependent components of WWBs. The authors should show the WWB statistics in the historical and future period in CESM-LENS.

The heat budget is calculated over a uniform 75-m mixed layer depth, but doesn't the mixed layer depth change under greenhouse forcing? It could be that changing it to e.g., 50-m won't make much difference, but it's worth a remark.

Finally, the writing should be improved. Often the points that the authors are trying to make are

not very accessible as they tend to follow description that is more supporting material, or more like a figure caption.

E.g., Opening of section 2, it takes three supplementary figures before going into the main Figure 1.

I find the paragraph in L85-101 breaks the flow. E.g., L85-89: These descriptions really belong to the figure caption of Fig. 1. L89-94: While noting model biases is necessary, it can be shortened and moved to Methods. Fig. 1a, e are identical, and Fig. 1 is very busy. I would suggest moving panel a, b, f, to a supplementary figure showing ERSSTv5, CESM20C, CMIP6 20C. Figure 1 then could just show panels c,d,g,h, and in fact this extra space can accommodate Figure S6c, d, g, h. I feel there are interesting results that are buried in the supplementary, e.g., Fig. S7. L199-201, 285-290 is another figure caption type of statement.

L38-41: this statement is not quite fitting here as the ocean thermostat mechanism would lead to La Nina-like mean state, while it's supposed to provide explanation on El Nino-like mean state in the preceding lines.

L46-52 does not flow very well. These two arguments are on opposing mechanisms yet leading to the same result, but the writing is such that the reader would expect an opposite result.

L102-109: The difference in SSTA shown in Fig. 1d, h, suggests that there appears to be an increased tendency for eastward evolution of the SST anomalies under greenhouse warming – an aspect covered previously by Santoso et al. (2013) tied to weakened Walker Circulation which is marked by eastward shift in convection, weaker Trade winds, currents.

Santoso, A. et al. Late-twentieth-century emergence of the El Niño propagation asymmetry and future projections. *Nature* 504, 126–130 (2013)

L114-115: It is stated that 'there is a consistent shift in the peak month from January to December' but this does not apply for the Nino3, both in CESM-LENS and CMIP6.

L133-134: point A is close to 120W and is thus in the eastern equatorial Pacific, not central as stated.

The description of Fig. S9 is fairly clear, but not so for Fig. S8: "changes in temperature and velocity contributions to the thermocline (left-column), zonal advective (middle-column)..." – are these simply the 21-20C change in say \bar{W} etc, and that these are ought to be compared to Fig. 2 bottom panels? Why does the change in dT'/dz (Fig. S8) look similar to the thermocline feedback term in Fig. 2 (bottom panel) but opposite sign?

L167-169 is contrary to L160-164. In L160-164 the increased Ekman feedback causes El Nino SSTA to persist longer in the eastern Pacific in late 21C, thus contributing to the eastward propagation. But in L167-169, the Ekman feedback tends to drag SSTAs westward. This needs to be clarified.

L249-250 on state dependent/independent WWBs, please add reference(s), e.g., Levine, A., Jin, F. F., & McPhaden, M. J. (2016) already in the reference list, and others.

L262-263: Please add reference supporting the statement on WWB change in CMIP5 models.

L91-292: '(not shown)'; it is shown in Cai et al. (2021)

In Concluding remarks, it will be good to discuss recent results from high-resolution models:
Future high-resolution El Niño/Southern Oscillation dynamics
Christian Wengel, Sun-Seon Lee, Malte F. Stuecker, Axel Timmermann, Jung-Eun Chu et al.
doi: 10.1038/s41558-021-01132-4

Robust decrease in El Niño/Southern Oscillation amplitude under long-term warming
Christopher W. Callahan, Chen Chen, Maria Rugenstein, Jonah Bloch-Johnson, Shuting Yang et al.
doi: 10.1038/s41558-021-01099-2

Reviewer #2 (Remarks to the Author):

This study investigates the impact of global warming on the temporal evolution of ENSO using CESM1 and CMIP6 model projections. The analyses are extensive and informative and the scientific subjects discussed are relevant to the ongoing interest in the ENSO research community to better understand the controlling factors of the evolution of ENSO. As I describe in the following comments, I have questions on some of authors' interpretations of the analysis results which may affect their conclusions. One of my main questions is related to their suggestion that there is a "tendency to have a shift towards stronger eastern Pacific El Niño events in the late 21C" (L170-171; also see my comment 9). Another major question I have is their discussion of the contribution of the Ekman feedback mechanism to the persistent warming of the 21C El Niño during its decaying phase. While I agree with the authors that the Ekman feedback is a responsible factor for the persistent decay warming, I have a different opinion on which period of the ENO lifecycle that this feedback mechanism should be emphasized for the explanation (see my comments 6-8 below). I suggest the authors add subtitles in all figure panels to help readers follow the discussions. Most of the figures have many panels in them. Without the subtitles, it is difficult to compare so many panels in one figure.

(1) Table S2: In this study, the duration of an ENSO event is expressed in days. I presume that the authors do this to aid their discussion of the WWB impact on ENSO. However, ENSO is a relatively slowly-varying climate phenomenon; I wonder if it is physically meaningful or reasonable to try to determine what day in a month that an ENSO event starts.

(2) L89-90: The CMIP6 ENSO (Fig. 1f) is weaker (rather than stronger) than the observed ENSO (Fig. 1e).

(3) L164-181 in SI: Please show EOF1 and EOF2 in the SI. The information would help readers to better understand why the four types of El Niño diversity can be defined by the formula stated in L178-181 without the need to read Lee et al. (2018).

(4) L126: The authors may want to consider revising this section title. Their discussion of Fig.1 concludes that the 21C El Niño is different from the 20C El Niño during their peak and decay phases. And the authors did not find consistent changes in the onset phase of the 21C and 20C El Niños in CESM-LENS and CMIP6. They stated that the "start date of the events, appears to be model dependent"(L114). Seems to me that the main question to be addressed in this section should be "What Drives the Projected Changes in El Niño During its PEAK (OR DEVELOPING) and Decay Phases?".

(5) L390; Eq (1): For the sake of completeness, please name each of the three RHS terms (i.e., the thermocline, zonal advective, and Ekman feedbacks) in this part of the SI.

(6) L136-142: I think the authors should place box B in a different part of the lower "Ekman" panel of Figure 2. The authors focused on the Ekman feedback term inside Box B to explain the decaying phase differences between the 21C and 20C El Niños. However, within Box B, this term produces positive heat content tendency differences in a region (between 160 and 100W) that is further away from the South American Coast, while the major heat content differences between 21C and 20C El Niños occurs in a region (between 130W-80W) immediately off the Coast (see the positive green contours during JAN-JUL of Year+1 in the lower Tendency panel of Fig. 2). Also, the Box B selected by the authors covers the period APR-JUL(+1), which is later than the period (JAN-JUL(+1)) when the positive ocean heat content differences (the green contours in the lower tendency panel of Fig. 2) and SSTAs (in Fig. 1d) start to appear in the eastern Pacific. From my own inspection of Figure 2, the Box B in the Ekman panel should be moved to the area where positive values of the Ekman term extending from 80W to 140W are found during JAN-MAR of Year(+1). The longitudinal and temporal locations of this new area match better with those of the positive heat content tendency (i.e., the red shadings) in the Tendency panel of Fig. 2.

It is important to clarify/determine which part of this Ekman feedback figure needs to be used to explain the persistent decaying of the 21C El Niño, because it affects the analysis and discussion of Figure S8 in L152-159.

(7) L156-159: Related to the previous comment, it seems to me that the W' -related enhancement of the Ekman feedback during boreal winter (Figure 9f) is more relevant than the mean stratification-related Ekman feedback during boreal spring (Figure 9i) to explain why the 21C El Niño can persist into boreal spring during its decaying phase.

(8) L161-162: Related to Comment (7), I cannot agree that “increased positive Ekman feedback between 150°W and 100°W) causes El Niño SSTAs to persist longer in the eastern Pacific in the late 21C”. I think it is the increased positive Ekman feedback between “ 120°W and 80°W ” during JAN-APR(+1) that should be emphasized to explain the persistent decaying feature. And this increased Ekman feedback can be attributed to the enhanced upwelling anomalies (W') of the 21C El Niño (Figure S9f, rather than Figure S9i). The following analysis should try to explain why the 21C El Niño can induce a stronger W' than the 20C El Niño.

(9) L170-171: I did not find any statements before this part of the manuscript that concluded from Figure 1 that there is a “tendency to have a shift towards stronger eastern Pacific El Niño events in the late 21C”. Figures 1d and 1h did suggest that the 21C El Niño has a stronger peak intensity than the 20C El Niño, but the increased SSTAs during Year(0) and the beginning of Year(+1) appear more in the central Pacific rather than in the eastern Pacific. Only during the decaying phase (i.e., Year +1) does the 21C El Niño linger and shift to the eastern Pacific.

(10) L187-188: Similar to the previous comment, are you referring to the decaying phase or the peak phase of the 21C El Niño? Please add the calendar months (e.g., JAN(0), ...) in these statements.

(11) L171-172: Please elaborate what you mean by “The delayed thermocline feedback..”. What is “delayed” referring to here?

(12) Figures S9 and S10: Please consider adding subtitles (or the term from Table S4) to each panel to make it easier to understand what contribution each panel represents. I have to admit that I have a difficult time to completely follow the discussions in L182-197. One main reason is because those discussions were based on the results presented in Figures S9 and S10, each of which include many panels that do not have subtitles. I have to go back and forth between these figures (and Figures 1 and 2) and figure captions (on the other pages of the manuscript) to remind myself what each panel means.

(13) Please consider using a few sentences to summarize the key findings from the heat budget analysis in Section 3.a.

(14) L291-292: What is the basis for the statement that the 21C El Niño “leading to a northeastward intensification of the circulation anomaly”?

(15) L296: Figure 4 reveals that the teleconnections are different between the 20C and 21C El Niños, but I don't see the reason why we can conclude that there is a “... projected enhancement of the 20C teleconnection into the 21C”. The spatial patterns of “21C minus 20C” do not resemble the spatial patterns of the 20C impacts.

(16) The remote impacts discussed in Section 4 are interesting but lack insights into the physical processes that can link the different evolution patterns of the 21C and 20C El Niños to their climate impacts over North and South America and Australia. The value of this section would be greatly increased if such insights were provided.

Reviewer #3 (Remarks to the Author):

Comments for “Projections of Faster Onset and Slower Decay of El Niño in the 21st Century” by Lopez et al.

Summary: This study investigates the change in seasonal evolution of El Niño under global

warming. Based on simulations by CESM and CMIP6 models, the authors report that El Niño is projected to develop more quickly in developing stage and persist longer under global warming. Furthermore, the change is considered to affect weather patterns around the globe.

Overall opinion: I think the results are interesting, but I think there are some problems for the paper. First, this study lacks a very clear and deep explanation or physical mechanism for the change. Second, this study lacks enough analyses about the uncertainty of the results, which reduce the confidence of conclusion. The specific comments for the two points are listed as follows.

Major comments

1. Although the authors do the mixed layer heat budget and think the change of Ekman feedback and thermocline feedback may be responsible for the change of El Niño seasonal evolution. But the reasons for the changes of some its components, for example $w^{\wedge'}$, $\partial T'/\partial Z$, WWBs, are not clearly given. In addition, I see a stronger cooling from Jan(1) to Apr(1) in the warming scenario from Fig. 2, which seems contrary to the authors' conclusion of slower dissipation of El Niño under global warming. Previous studies show that the interaction between eastern Pacific ITCZ and El Niño in boreal spring is key to determine El Niño decaying speed. Does the interaction contribute to the slow decay of El Niño under global warming?

2. In this study, the authors use Student's t-test to show the confidence level of their results. Such analyses may be not enough to show that their results are confident. I recommend the author do some further analyses about the uncertainty of the results among different models of members, and give reasons for the uncertainty.

Please extend our gratitude to the three anonymous reviewers for their constructive comments/suggestions that have led to a significant improvement to our manuscript.

The following overarching changes have been made to the manuscript in order to address the reviewers' comments/suggestions:

- Clarify the language and thus set apart the major contributions of this work from previous studies.
- Expanded the heat budget analysis to discuss nonlinear ENSO processes which have been shown to be important for ENSO evolution.
- Improved section 4 (i.e., remote impacts) by focusing on the boreal fall (i.e., developing) and boreal spring (i.e., decay) periods to highlight the impact of faster growth and slow dissipating of the future El Niño events for North America, South America, and Australia.
- Added a more robust statistical significance test (i.e., Monte Carlo bootstrapping, see Methods) that is non parametric and as such does not make assumptions on the distribution of the data. This is possible given the large ensemble size analyzed here.
- Added subtitles in all figure panels, including the supplementary figures, to help readers follow the discussions.

In what follows, our responses are highlighted in boldface, while the original reviewers' suggestions are in normal font.

Reviewer #1 (Remarks to the Author):

This is a nice study that investigates the response of El Nino to greenhouse forcing with a focus on the ENSO onset and decay phases. The issue is an important one to resolve given the significant impact of El Nino on climate, and the ways El Nino will change in a warming planet need to be understood. The study found that El Nino is projected to grow faster and persist longer over the eastern Pacific with significant remote impacts in air temperature and precipitation over the American continent and Australia. The analysis is quite thorough and comprehensive, but some clarifications are required and the writing can be improved, along with additional of relevant references for a more balanced paper.

A main critique would be a lack/omission of discussion on nonlinear ENSO processes which have been shown to be important for understanding the response to greenhouse forcing, e.g., see a review paper that recently came out: Cai et al. (2021) Nature Reviews Earth and Environment, <http://dx.doi.org/10.1038/s43017-021-00199-z>.

In this present analysis, the focus is on the linear terms, while the nonlinear terms are lumped together in a residual, along with air-sea heat fluxes. This may obscure their role in the different phases.

Response: We agree with the reviewer and now incorporated further analysis/discussion on the role of the nonlinear ENSO processes in the differences in spatio-temporal El Niño evolution. Similar to Fig. 2 in the original paper, Fig. R1 below shows the decomposition of the residual term of the heat budget into the contribution from surface heat fluxes and nonlinear advection terms. Note that the residual term still comprises diffusive heating, unresolved sub-grid scale heat fluxes, and sub-monthly scale heating. Overall, the contribution of the nonlinear terms to the heat budget analysis is much smaller in amplitude than those of the main ENSO feedbacks (e.g., thermocline, zonal advective, and Ekman feedback terms, Fig. 2). However, a projected increase in the nonlinear meridional advection may support the growth of future El Niño events (Fig. R1, label A) whereas an increase in the nonlinear zonal advection might aid in the persistence of future El Niño events into the boreal spring in the eastern Pacific (Fig. R1, label B). It should also be noted that the total residual term (left figure of Fig. R1) damps the heat content anomaly (Fig. 2), and its damping role is projected to increase further in the 21C.

We have added this analysis and Fig. R1 below to the revised manuscript as Supplementary Fig. 4.

Figure R1. Similar to Fig. 2 but showing the residual terms of the mixed layer heat budget analysis during El Niño events for CESM-LENS ($W m^2$). The top row shows the composites for the residual, air-sea net heat fluxes, and the three nonlinear advective terms (i.e., zonal, meridional, and vertical advection). Bottom row shows the projected future change in the composite, for the 21st minus the 20th Century (21C). All terms are computed assuming a constant 75 m mixed layer depth. Only anomalies that exceed the 95% confidence level, based on a bootstrapping technique, are shown. See text for references to points A and B.

How well models simulate ENSO nonlinearity has also been deemed to be important. The inter-model consensus for increased ENSO SST variability has been based on models that reasonably simulate ENSO nonlinearity, and thus model selection is necessary. This was the case for CMIP5 models (Cai et al. 2018), but for CMIP6 it turns out that model selection is not quite necessary as ENSO nonlinearity is quite reasonable in these models (Cai et al. 2021). This is worth to be discussed.

Response: Thanks for the suggestion. We agree that model selection was unfortunately an essential part of describing ENSO variability, and future projections. Model biases often obscure and make distinction between natural variability and external forcing difficult. This is one of the main reasons why we employ the CESM-Large Ensemble (CESM-LENS), as internal variability can be readily extracted as the difference between an individual ensemble member and the ensemble mean. However, in this way, the ensemble mean contains the model bias. This further motivated us to look at more state-of-the-art models (i.e., CMIP6). Cai et al. 2021 discuss many shortcomings of simulated nonlinearities etc. in their concluding section. And it's possible that most/all of the models have strong biases in nonlinearities, since these nonlinearities haven't been well observed.

We have now added discussion on model selection and added reference to Cai et al. 2018 and 2021 to the revised manuscript Method section.

There are studies that have concluded El Niño to La Niña transition being enhanced under greenhouse warming (e.g., An & Kim 2018, <https://agupubs.onlinelibrary.wiley.com/doi/full/10.1029/2018GL078476>). How to reconcile such result with the prolonged El Niño found in this present study?

Response: The results of An & Kim 2018 addressed both, the internal and external causes of asymmetry in the ENSO transition. They found that under the more extreme emission scenario (i.e., RCP8.5), there is a significant increase in the El Niño-to-La Niña transition. While we did not look at El Niño-to-La Niña transition, we found an increase in the frequency of occurrence of El Niño events in both CESM-LENS and CMIP6 (Supplementary Fig. 2). This increased occurrence of El Niño in the 21st Century coupled to the reported larger amplitude and prolonged El Niño events (Fig. 1) could increase the transition of the El Niño-to-La Niña. This is reflected in Fig. R2 below, where there is a projected increase in the number of El Niño and La Niña events (Fig. R2 panel a) as well as El Niño-to-2 year La Niña transition (Fig. R2 panel b). This is consistent with DiNezio et al. 2017, which found that a strong thermocline discharge or a strong El Niño can lead to La Niña conditions that last 2 years. Also note that we report in this work a projected increase in the negative thermocline feedback over the central Pacific during the decay phase (Fig. 2 point labeled C), as well as increased negative SSTA anomalies following the peak El Niño events (Fig. 1), suggesting an enhanced El Niño-to-La Niña transition, as reported in An & Kim 2017.

While El Niño-to-La Niña transition is an important aspect of future ENSO changes, analysis regarding La Niña events is outside the main scope of this work and will be further investigated in a separate study. However, we briefly added the discussion above to the

revised manuscript, reconciling the findings of An and Kim (2018) to our findings here in section 2 of the revised paper.

An, S. I., & Kim, J. W. (2018). ENSO transition asymmetry: Internal and external causes and intermodel diversity. *Geophysical Research Letters*, 45(10), 5095-5104.

DiNezio, P. N., Deser, C., Okumura, Y., & Karspeck, A. (2017). Predictability of 2-year La Niña events in a coupled general circulation model. *Climate dynamics*, 49(11), 4237-4261.

Figure R2. a) Temporal evolution of a 50-year running averaged number of El Niño (red) and La Niña (blue) events from CESM-LENS simulations (blue). The boxes denote the interquartile range, and the whiskers denote 5th-95th percentile range, of event counts estimated by randomly selecting 15 out of the 30 ensemble members 1000 times and repeating the event count. The light-colored horizontal interval denotes the natural variability range, computed from the interquartile range from a 1100-year pre-industrial simulation of CESM-LENS by randomly selecting 50-year periods for the event count. The year labels on the abscissa correspond to the central year of the 50-year window. For example, the year 2070 indicates the period spanning 2046-2095. The 21C projections are from the RCP8.5 from CESM-LENS.

Section 4 on El Niño impacts is rather short and rudimentary relative to the number of figures that they take up in the manuscript (50% of total figures). There's ought to be more discussions on how ENSO impacts on these regions with the relevant literature cited, including those that used CMIP5 models, e.g., Grose et al. (2020) for

Australia <https://agupubs.onlinelibrary.wiley.com/doi/full/10.1029/2019EF001469>; Cai et al. 2020 on South America rainfall and references therein <https://www.nature.com/articles/s43017-020-0040-3>;

Response: We agree that section 4 needs more succinct description of the remote impacts and literature review on previous work. As such we added the discussion below on previous findings and condensed the results by showing results from CESM-LENS (Fig.s 4 and 5) and moving the CMIP6 results to the supplementary material (Supplementary Fig.s 11 and 12).

The El Niño-driven circulation changes, as measured by the composite of 500 hPa geopotential height, is dominated by the Pacific South American pattern during the developing phase (Rodrigues et al. 2015) (Supplementary Fig. 10a) and the Pacific North American pattern during the decay phase (Trenberth et al. 1998) (Supplementary Fig. 10c). These patterns are primarily driven by the upper tropospheric divergence flow associated with tropical convection from El Niño.

Note that the tropical Pacific atmospheric forcing is projected to increase in the 21C relative to the 20C for the growth (Fig. R3, which is Supplementary Fig. 10 of the revised paper) and demise phase (Fig. R3), leading to an intensification of the circulation anomaly. In addition, there is an eastward shift of the center velocity potential pattern in the tropics from 20C to 21C, consistent with Cai et al. 2021. The projected enhancement of the 20C teleconnection into the 21C is consistent with previous work, which showed that sufficiently warm and persistent SSTA in the far eastern equatorial Pacific is required to excite teleconnection patterns (Lee et al. 2018). Future changes in El Niño's teleconnections during the developing phase (SON in Year 0) are mostly over the Southern Hemisphere (Fig. R3b). This is consistent with enhanced SSTAs and precipitation over the tropical Pacific (Fig. 1), the associated upper tropospheric teleconnection response, and the notion that the Southern Hemisphere ENSO response leads the ENSO peak in the tropics. During the decay phase (MAM in Year +1, Fig. R3b), El Niño's remote effects mimic those of the peak phase, with enhanced future teleconnections.

South America:

ENSO influences South American climate through modulations of the Walker circulation as well as extra-tropical teleconnections (e.g., Rossby wave trans, Ropelewski et al. 2008, Cai et al. 2020), producing a north-south dipole pattern in temperature and rainfall with cooler and wetter (warmer and drier) conditions in the Southern (Northern) portion of the continent. In addition, remote ENSO effects over SA strongly depends not only on amplitude but also the longitudinal location of maximum SSTA, with EP events exhibiting more pronounced shift in the Walker Circulation (Tedeschi et al. 2016) and stronger extratropical teleconnections (Rodrigues et al. 2015 and thus impacts on SA than CP events (Cai et al. 2020). ENSO impact over South America is also projected to increase in the future, with a projected increase (decrease) rainfall over the Southeastern South America (Amazon basin) (Cai et al. 2020).

Australia:

ENSO is negatively correlated with rainfall, especially over the north and eastern portions of Australia (Cai et al. 2011). El Niño events are associated with warming and reduced precipitation over Australia during the historical period (Fasullo). However, typical ENSO

response over Australia varies significantly with the location of maximum SSTA. For instance, central Pacific events tend to produce more widespread precipitation and temperature signal in Australia than those of the eastern Pacific type (Capotondi et al. 2015).

Note that we added Fig. R3 below to the supplementary material (as Supplementary Fig. 10) depicting the circulation changes associated with El Niño events and the future projected changes for the developing (SON) and decay phases (MAM) to the revised text.

Figure R3. a) Composite analysis of 200hPa velocity potential (color, interval 10^6 s^{-1}) and 500hPa geopotential height (contour, interval 5 m) anomalies during El Niño events for September-October-November (SON, or developing year-0) for late 20th Century and b) the projected changes (i.e., 21C minus 20C) of 200hPa velocity potential (color, interval 10^5 s^{-1}) and 500hPa geopotential height (contour, interval 2 m). Panels c) and d) are similar to a) and b) respectively but for the March-April-May (MAM, or decay year+1). See Methods for Definition of El Niño.

Cai, W., Van Rensch, P., Cowan, T., & Hendon, H. H. (2011). Teleconnection pathways of ENSO and the IOD and the mechanisms for impacts on Australian rainfall. *Journal of Climate*, 24(15), 3910-3923.

Cai, W., McPhaden, M. J., Grimm, A. M., Rodrigues, R. R., Taschetto, A. S., Garreaud, R. D., ... & Vera, C. (2020). Climate impacts of the El Niño–southern oscillation on South America. *Nature Reviews Earth & Environment*, 1(4), 215-231.

Capotondi, A., Wittenberg, A. T., Newman, M., di Lorenzo, E., Yu, J. Y., Braconnot, P., ...

- Yeh, S. W. (2015). Understanding ENSO diversity. *Bulletin of the American Meteorological Society*, 96(6), 921–938. <https://doi.org/10.1175/BAMS-D-13-00117.1>
- Lee, S. K., Lopez, H., Chung, E. S., DiNezio, P., Yeh, S. W., & Wittenberg, A. T. (2018). On the fragile relationship between El Niño and California rainfall. *Geophysical Research Letters*, 45(2), 907–915. <https://doi.org/10.1002/2017GL076197>
- Rodrigues, R. R., Campos, E. J. D. & Haarsma, R. The impact of ENSO on the South Atlantic subtropical dipole mode. *J. Clim.* 28, 2691–2705 (2015).
- Ropelewski, C. F. & Bell, M. A. Shifts in the statistics of daily rainfall in South America conditional on ENSO phase. *J. Clim.* 21, 849–865 (2008).
- Tedeschi, R. G., Grimm, A. M. & Cavalcanti, I. F. Influence of Central and East ENSO on precipitation and its extreme events in South America during austral autumn and winter. *Int. J. Climatol.* 36, 4797–4814 (2016).
- Trenberth, K. E., Branstator, G. W., Karoly, D., Kumar, A., Lau, N. C., & Ropelewski, C. (1998). Progress during TOGA in understanding and modeling global teleconnections associated with tropical sea surface temperatures. *Journal of Geophysical Research: Oceans*, 103(C7), 14291–14324.

How the anomalies are obtained is not clear. For the CESM-LENS, it is understood that the ensemble mean can be subtracted from each member. But for the CMIP6 models, how is this done when every model has different climate sensitivity?

Response: We have now clarified how the anomalies are obtained for CESM-LENS and CMIP6 in the method sections of the revised paper as well as added Fig. R4 and discussion below to the Supplementary Fig. 16 of the revised text.

For CESM-LENS (Method#1, here forward), the ensemble mean is subtracted for each ensemble member, then a monthly climatology is further removed from the resultant anomalies. This is done to avoid the potential for each ensemble member having a slightly different climatology. This method, should take care of any trend (including non-linear) due to external forcing as well as CESM model bias.

For CMIP6 (Method#2, here forward), and as the reviewer pointed out, each model has its own physics and thus bias, so the ensemble mean approach is not applicable. To compute anomalies in CMIP6, we remove a 30-year running mean climatology, similar to what is done at NOAA/CPC to identify ENSO events.

We have verified these two methods of defining anomalies in a trending climate. Figure R1 (panel a) shows the Niño3.4 SSTA timeseries reconstructed using the two method for a randomly selected ensemble member. Figure R1 (panel b) shows a Taylor diagram from all 30 CESM-LENS ensembles computed using both methodologies outlined above. Note that the temporal correlation between the two methods is higher than 0.95 for all ensembles, with root mean square errors smaller than 35 percent of the standard deviation of Niño3.4 ($RMSE < 0.35\sigma$). The two methods yield nearly identical results. However, only Method#2 is appropriate for use with multi-ensemble simulations of different models (e.g., CMIP6), while both methods are appropriate for multi-ensemble simulations of a single model (e.g., CESM-LENS). Method#1 was chosen for CESM-LENS as it is significantly less

computational demanding, especially when quantifying multi-dimensional anomalies (e.g., sub-surface temperature and velocities) as a function of latitude, longitude, depth, time, and ensemble member.

Figure R4. a) Niño3.4 SSTA timeseries for a randomly chosen ensemble member of the CESM-LENS computed from the deviation from the ensemble mean (i.e., Method#1, red) and from the deviation from a 30-year running average (i.e., Method#2, blue). b) Taylor diagram depicting the standard deviation (abscissa), correlation (azimuth), and root mean square error (concentric circles) of the two methods of defining Niño3.4 SSTA anomalies for all 30 ensembles from the CESM-LENS. All timeseries were normalized based on the standard deviation from Method#1 (red dot) to facilitate comparison against Method#2 (blue dots).

Please indicate statistical significance in Figure 2. In the caption, mention also it is longitude-time for the average over 5S-5N. There are other figures that do not have statistical significance prescribed (e.g., Fig. S8, S9).

Response: We have now added a statistical significance to Fig. 2 and other plots of the revised paper. Please note that due to Fig. 2 being a busy multi-panel plot, we are now only showing the composite anomalies that exceeded the significance level. All values below the significance levels are masked out. Also note, that throughout the paper, we now relied on a non-parametric bootstrapping technique to assess statistical significance, rather than a student-T test. Please refer to our answer to Reviewer#3 comments for more thoroughly

description of the statistical significance method. We have also clarified this in the revised manuscript.

L243-257: there seems to be no figure that explicitly shows changes in WWB activity, and no analysis that supports the statement on the enhancement of the stochastic and state-dependent components of WWBs. The authors should show the WWB statistics in the historical and future period in CESM-LENS.

Response: Please note that Fig. 3g depicts a snapshot to show the relationship between a zonal wind stress forcing of the form of WWBs and the associated Kelvin wave response as outline by equation 2 and in Zhang and Gottschalck, 2002 (Ref. 104 in the revised paper). Note that the conclusion of increased WWBs activity and thus downwelling Kelvin wave is assessed by the plot in Fig. 3h, which show the changes in WWBs-induced downwelling Kelvin waves as a function of calendar month in the 20C and 21C periods separately. We can observe a statistically significant increase in WWB activity in the 21C (red line) compared to that of 20C (blue line).

We have clarified this in the revised text and Fig. 3 caption.

Zhang, C., & Gottschalck, J. (2002). SST anomalies of ENSO and the Madden–Julian oscillation in the equatorial Pacific. *Journal of Climate*, 15(17), 2429-2445.

The heat budget is calculated over a uniform 75-m mixed layer depth, but doesn't the mixed layer depth change under greenhouse forcing? It could be that changing it to e.g., 50-m won't make much difference, but it's worth a remark.

Response: Agree, we did test different mixed layer depths ranging from 30m to 100m and found that the main conclusion is consistent and independent on the mixed layer depth. This is noted in the Mixed Layer Heat Budget subsection of the Methods section of the revised paper.

Finally, the writing should be improved. Often the points that the authors are trying to make are not very accessible as they tend to follow description that is more supporting material, or more like a figure caption.

E.g., Opening of section 2, it takes three supplementary figures before going into the main Figure 1.

Response: We agree, and as such, we have now refined the language to more succinctly convey the main results, and avoiding unnecessary description and references to supporting material in the main text.

I find the paragraph in L85-101 breaks the flow. E.g., L85-89: These descriptions really belong to the figure caption of Fig. 1.

Response: Agree, we removed some of the redundant language in the main text and expanded the figure captions.

L89-94: While noting model biases is necessary, it can be shortened and moved to Methods.

Response: We moved the model bias discussion to the Methods section as well as to the Supplementary Information.

Fig. 1a, e are identical, and Fig. 1 is very busy. I would suggest moving panel a, b, f, to a supplementary figure showing ERSSTv5, CESM20C, CMIP6 20C. Figure 1 then could just show panels c,d,g,h, and in fact this extra space can accommodate Figure S6c, d, g, h. I feel there are interesting results that are buried in the supplementary, e.g., Fig. S7.

Response: This is a great suggestion. As the reviewer suggested, we have now modified Fig. 1 to include the precipitation changes, which were previously hidden in the supplementary material. The new Fig. 1 now combines SSTA and precipitation anomalies for the 21C and the difference between 21C minus 20C. Comparison among ERSSTv5, CESM20C, CMIP6 20C were moved to the supplementary Fig. 15 as recommended.

L199-201, 285-290 is another figure caption type of statement.

Response: We shortened the figure description in the revised manuscript, relegating details to the figure captions.

L38-41: this statement is not quite fitting here as the ocean thermostat mechanism would lead to La Nina-like mean state, while it's supposed to provide explanation on El Nino-like mean state in the preceding lines.

Response: We modified this statement to better highlight the different mechanisms for future changes in the mean state.

L46-52 does not flow very well. These two arguments are on opposing mechanisms yet leading to the same result, but the writing is such that the reader would expect an opposite result.

Response: Agree, We have modified these two arguments to show that the two opposing mechanisms (i.e., El Niño-like versus La Niña-like mean state changes) could lead to the same result (i.e., increase the likelihood of extreme El Niño events).

L102-109: The difference in SSTA shown in Fig. 1d, h, suggests that there appears to be an increased tendency for eastward evolution of the SST anomalies under greenhouse warming – an aspect covered previously by Santoso et al. (2013) tied to weakened Walker Circulation which is marked by eastward shift in convection, weaker Trade winds, currents.

Santoso, A. et al. Late-twentieth-century emergence of the El Niño propagation asymmetry and future projections. *Nature* 504, 126–130 (2013)

Response: We concur, the projected enhanced (reduced) SSTA in the eastern (central) Pacific is consistent with an increased eastward evolution of SSTA, both in CESM-LENS

and CMIP6 composites. Therefore, we have noted these changes in the revised text as well as added citation for Santoso et al. 2013; Chen et al. 2017.

Chen, C., Cane, M. A., Wittenberg, A. T., & Chen, D. (2017). ENSO in the CMIP5 simulations: life cycles, diversity, and responses to climate change. *Journal of Climate*, 30(2), 775-801.

Santoso, A. et al. Late-twentieth-century emergence of the El Niño propagation asymmetry and future projections. *Nature* 504, 126–130 (2013)

L114-115: It is stated that ‘there is a consistent shift in the peak month from January to December’ but this does not apply for the Niño3, both in CESM-LENS and CMIP6.

Response: Agree, we have modified the statement specifically to Niño4 and Niño3.4 regions only.

L133-134: point A is close to 120W and is thus in the eastern equatorial Pacific, not central as stated.

Response: Agree, we change this distinction to “eastern Pacific” in the revised text.

The description of Fig. S9 is fairly clear, but not so for Fig. S8: “changes in temperature and velocity contributions to the thermocline (left-column), zonal advective (middle-column)...” – are these simply the 21-20C change in say \bar{w} etc, and that these are ought to be compared to Fig. 2 bottom panels?

Response: Please note that former Supplementary Fig. S8 is now Supplementary Fig. 5 and former Supplementary Fig. S9 is now Supplementary Fig. 6, as the reviewer asserted, is simply each of the feedback terms decomposed into their components (e.g., \bar{w} and $\frac{\partial T'}{\partial z}$ for the thermocline feedback, u' and $\frac{\partial \bar{T}}{\partial x}$ for the zonal advective feedback, and w' and $\frac{\partial \bar{T}}{\partial z}$ for the Ekman feedback). They are intended to be compared to Fig. 2 (bottom panels), which is repeated in Fig. S9 top row (now Supplementary Fig. 6). This is done to facilitate the discussion of relating changes in the mean state versus changes in the anomalies presented in Supplementary Fig. 6. We have added more description of these figures in the figure captions as well as more subtitles to each of the panels for easier reference.

Why does the change in dT'/dz (Fig. S8) look similar to the thermocline feedback term in Fig. 2 (bottom panel) but opposite sign?

Response: Please note that former Supplementary Fig. S8 is now Supplementary Fig. 5. Regarding the sign of $\partial T'/\partial z$ in Supplementary Fig. 5, it was chosen to be consistent with the heat budget equation (eq. 1), given that there is a negative sign in front of the integrand comprising all three major feedback terms (i.e., thermocline, zonal advective, and Ekman). Note that a negative $\partial T'/\partial z$ will act as positive feedback to $\partial T'/\partial t$ if \bar{w} is positive (i.e., upwelling). The sign convention was chosen this way given the non-linearity aspects of equation 1.

This is clarified in the revised manuscript and in the supplementary material.

L167-169 is contrary to L160-164. In L160-164 the increased Ekman feedback causes El Niño SSTA to persist longer in the eastern Pacific in late 21C, thus contributing to the eastward propagation. But in L167-169, the Ekman feedback tends to drag SSTAs westward. This needs to be clarified.

Response: The statement in L160-164 regarding the increased Ekman feedbacks relates to the projected changes in the Ekman feedback during 21 Century relative to the 20 Century, which is labeled point B in Fig. 2 and shows a positive feedback anomaly (i.e., reinforcing the positive SSTA) during the termination of El Niño. In contrast, the statement in L167-169, relating the Ekman Feedbacks is in relation to the 20C, which shows a negative feedback anomaly (i.e., damping of the positive SSTA; top row in Fig. 2). Thus, while the Ekman feedback tends to drag SSTA evolution westward (e.g., negative feedback in the far eastern Pacific during the 20C), the projected changes in the Ekman feedback is a positive anomaly in the eastern Pacific. This coupled to an enhanced thermocline feedback cause El Niño SSTAs to persist longer in the eastern Pacific in the late 21C relative to the 20C.

With that said, we acknowledge that the original statement was not well constructed. We have clarified this statement in the revised manuscript.

L249-250 on state dependent/independent WWBs, please add reference(s), e.g., Levine, A., Jin, F. F., & McPhaden, M. J. (2016) already in the reference list, and others.

Response: References for studies addressing the distinction between state-independent and state-dependent WWBs are now added to the statement. These references were already listed in the original text (e.g., Gebbie et al. 2007; Lopez et al. 2013; Lopez and Kirtman 2013; Levine et al. 2016).

L262-263: Please add reference supporting the statement on WWB change in CMIP5 models.

Response: We added a reference in support of the WWB changes in CMIP5 (Levine et al. 2016).

L91-292: '(not shown)'; it is shown in Cai et al. (2021)

Response: We added the citation for Cai et al. 2021 in the revised manuscript.

In Concluding remarks, it will be good to discuss recent results from high-resolution models: Future high-resolution El Niño/Southern Oscillation dynamics:

Christian Wengel, Sun-Seon Lee, Malte F. Stuecker, Axel Timmermann, Jung-Eun Chu et al.
doi:10.1038/s41558-021-01132-4

Robust decrease in El Niño/Southern Oscillation amplitude under long-term warming

Christopher W. Callahan, Chen Chen, Maria Rugenstein, Jonah Bloch-Johnson, Shuting Yang et al.
doi:10.1038/s41558-021-01099-2

Response:

We added discussion on recent results from Wengel et al. 2021 and Callahan et al. 2021 in the revised manuscript.

A recent study analyzed the transient and equilibrium response of ENSO to anthropogenic forcing under millennial-scale warming in high emission scenarios from Long Run Model Intercomparison (Callahan et al. 2021). They found that while the transient response to anthropogenic forcing shows large uncertainties, the equilibrium response manifests a decreased in ENSO amplitude and no change in ENSO frequency. In addition, the observed trend in Pacific mean state changes is well within the unforced variability range. However, it is still unclear whether current models can reproduce ENSO changes on millennial timescales.

Another limitation of future projections is the difficulties representing sub-grid scale processes and feedbacks (atmospheric convection and clouds, and near-surface ocean mixing) in low-resolution models such as those from CMIP, which may affect model projections of future ENSO behavior. A recent study found that resolving fundamental mesoscale features produce a contrasting result when compared to low-resolution ENSO projections. That is, anthropogenic forcing induces a weakening of future ENSO variability (Wengel et al. 2021). It is noted that mesoscale features such as tropical instability waves, which are not resolved in low-resolution models, serve as an equally important damping mechanism for ENSO on the same order of magnitude as to the thermodynamic and dynamic damping terms. However, as pointed out in Wengel et al. 2021, there are still significant model biases in eddy-resolving simulations, as well as important processes which are still parametrized even in high-resolution simulations. This coupled to the limited simulation length of 100 years and the lack of multi-model ensemble call for the need for further studies to confirm the contrasting results of ENSO projections between low- and high-resolution simulations.

Callahan, C. W., Chen, C., Rugenstein, M., Bloch-Johnson, J., Yang, S., & Moyer, E. J. (2021). Robust decrease in El Niño/Southern Oscillation amplitude under long-term warming. *Nature Climate Change*, 11(9), 752-757.

Wengel, C., Lee, S. S., Stuecker, M. F., Timmermann, A., Chu, J. E., & Schloesser, F. (2021). Future high-resolution El Niño/Southern Oscillation dynamics. *Nature Climate Change*, 11(9), 758-765.

Reviewer #2 (Remarks to the Author):

This study investigates the impact of global warming on the temporal evolution of ENSO using CESM1 and CMIP6 model projections. The analyses are extensive and informative and the scientific subjects discussed are relevant to the ongoing interest in the ENSO research community to better understand the controlling factors of the evolution of ENSO. As I describe in the following comments, I have questions on some of authors' interpretations of the analysis results which may affect their conclusions. One of my main questions is related to their suggestion that there is a "tendency to have a shift towards stronger eastern Pacific El Niño events in the late 21C" (L170-171; also see my comment 9). Another major question I have is their discussion of the contribution of the Ekman feedback mechanism to the persistent warming of the 21C El Niño during its decaying phase. While I agree with the authors that the Ekman feedback is a responsible factor for the persistent decay warming, I have a different opinion on which period of the ENO lifecycle that this feedback mechanism should be emphasized for the explanation (see my comments 6-8 below). I suggest the authors add subtitles in all figure panels to help readers follow the discussions. Most of the figures have many panels in them. Without the subtitles, it is difficult to compare so many panels in one figure.

Response: We are very grateful for your constructive comments/suggestions and hope to have successfully addressed all your inquiries. Also note that we added subtitles in all figure panels, including the supplementary figures, to help readers follow the discussions.

Please find our replies to your specific comments below in bold font.

(1) Table S2: In this study, the duration of an ENSO event is expressed in days. I presume that the authors do this to aid their discussion of the WWB impact on ENSO. However, ENSO is a relatively slowly-varying climate phenomenon; I wonder if it is physically meaningful or reasonable to try to determine what day in a month that an ENSO event starts.

Response: We concur that it is relatively arbitrary to reference the duration of ENSO events based on specific day of the month. We were following the defining onset, duration, and demise based on Fang et al. 2020 in order to highlight the differences in duration of the events as well as the model ensemble spreads or uncertainty ranges. If we report the start and end in term of just the month, we would miss some of the significant changes from the 20C to the 21C as well as the spread among CESM and CMIP6 ensembles. Start and end day ranges among ensemble members, as shown in Supplementary Tables 2 and 3, are more meaningful than pinpointing an exact date for such a slow-varying phenomenon as you suggested.

Fang, S. W., & Yu, J. Y. (2020). Contrasting transition complexity between El Niño and La Niña: Observations and CMIP5/6 models. *Geophysical Research Letters*, 47(16), e2020GL088926.

(2) L89-90: The CMIP6 ENSO (Fig. 1f) is weaker (rather than stronger) than the observed ENSO (Fig. 1e).

Response: Agree, the CMIP6 ensemble mean is weaker than observed, whereas the CESM-LENS has a stronger amplitude. This is also consistent with the Taylor diagram (supplementary Figure 14). We have corrected this in the revised text (note that this model comparison with observation has been moved to the Method section).

(3) L164-181 in SI: Please show EOF1 and EOF2 in the SI. The information would help readers to better understand why the four types of El Niño diversity can be defined by the formula stated in L178-181 without the need to read Lee et al. (2018).

Response: We added EOF1 and EOF2 to the now supplementary Fig. 3 in the revised manuscript. Note that we also added the composite mean (Fig. R5 a) for reference.

Figure R5. Spatio-temporal El Niño diversity from CESM-LENS 20C expressed from the combination of the two leading empirical orthogonal functions. a) composite mean, b) EOF1, and c) EOF2 of the longitude-time evolution of SSTA. d) SSTA evolution of the transitioning El Niño flavor (e.g., composite mean + EOF1), e) resurgent flavor (e.g., composite mean - EOF1), f) persistent flavor (e.g., composite mean + EOF2), and g) early-terminating flavor (e.g., composite mean - EOF2). Panel h) shows the 20C (blue) and 21C (red) bi-variate distribution of the phase-space relationship between principal components (PC1 and PC2). The percentage values in parentheses indicate the projected percentage increase in the

specific event flavor in the 21C relative to the 20C. The blue (red) values indicate the percentage of events in each category out of the total events for the 20C (21C).

(4) L126: The authors may want to consider revising this section title. Their discussion of Fig.1 concludes that the 21C El Nino is different from the 20C El Nino during their peak and decay phases. And the authors did not find consistent changes in the onset phase of the 21C and 20C El Ninos in CESM-LENS and CMIP6. They stated that the “start date of the events, appears to be model dependent”(L114). Seems to me that the main question to be addressed in this section should be “What Drives the Projected Changes in El Niño During its PEAK (OR DEVELOPING) and Decay Phases?”.

Response: We acknowledge that more clarification is needed in this section. Our findings indicate that El Niño evolution is different in both, the developing period e.g., September-November (year 0) as well as the decay period e.g., March-May (year +1). For example, while the start date is model dependent as you pointed out, the 21C growth rate is significantly larger than the 20C with distinct spatio-temporal evolution in both CMIP6 and CESM-LENS (Table S2 and S2). This coupled to the distinct spatio-temporal evolution of the SSTA and precipitation anomalies (Fig. 1) as well as the dominant feedback processes discussed in section 2 (Fig. 2) shows that future El Niño events are projected to grow at a faster rate. In order to clarify this, we have added more references to the text when referring to specific figures (e.g., by pointing at specific locations in the figures). We have also changed the section title as suggested to “What Drives the Projected Changes in El Niño During its developing and Decay Phases?”.

(5) L390; Eq (1): For the sake of completeness, please name each of the three RHS terms (i.e., the thermocline, zonal advective, and Ekman feedbacks) in this part of the SI.

Response: We have now named all terms in the revised text.

(6) L136-142: I think the authors should place box B in a different part of the lower “Ekman” panel of Figure 2. The authors focused on the Ekman feedback term inside Box B to explain the decaying phase differences between the 21C and 20C El Ninos. However, within Box B, this term produces positive heat content tendency differences in a region (between 160 and 100W) that is further away from the South American Coast, while the major heat content differences between 21C and 20C El Ninos occurs in a region (between 130W-80W) immediately off the Coast (see the positive green contours during JAN-JUL of Year+1 in the lower Tendency panel of Fig. 2). Also, the Box B selected by the authors covers the period APR-JUL(+1), which is later than the period (JAN-JUL(+1)) when the positive ocean heat content differences (the green contours in the lower tendency panel of Fig. 2) and SSTAs (in Fig. 1d) start to appear in the eastern Pacific. From my own inspection of Figure 2, the Box B in the Ekman panel should be moved to the area where positive values of the Ekman term extending from 80W to 140W are found during JAN-MAR of Year(+1). The longitudinal and temporal locations of this new area match better with those of the positive heat content tendency (i.e., the red shadings) in the Tendency panel of Fig. 2.

Response: We agree, the box label B in the Ekman feedback difference plot is now moved to the region comprised between 130W-80W and JAN-MAR (year +1). This one does

indeed explain the tendency in the heat content and its subsequent temporal evolution (eq. R1 below). Also note that we added statistical significance test to the revised Fig. 2 and are only showing composites that are significant based on a Monte Carlo subsampling technique. (Please refer to the revised manuscript and/or answer to the 3rd Reviewer last question for more details on the statistical significance text).

$$\int_{-H}^0 \frac{\partial T'}{\partial t} dz \sim - \int_{-H}^0 \left[w' \frac{\partial \bar{T}}{\partial z} \right] dz \quad R1$$

It is important to clarify/determine which part of this Ekman feedback figure needs to be used to explain the persistent decaying of the 21C El Nino, because it affects the analysis and discussion of Figure S8 in L152-159.

Response: Please note that supplementary figures 8 and 9 are now supplementary Figs. 5 and 6 in the revised text respectively.

Note that the placing of box B on Fig. 2 does not affect the findings reported that the projected increase of the Ekman feedback in the eastern Pacific between 130°W-80°W and JAN-MAR (year +1) is well explained by the contribution from anomalous upwelling (w' , Supplementary Fig. 5 and Fig. 6f) and an increase in the mean stratification ($\frac{\partial \bar{T}}{\partial z}$, Supplementary Fig. 5 and Fig. 6i). As we noted in the paper, the largest changes in w' is mostly in the central Pacific; however, $\frac{\partial \bar{T}}{\partial z}$ dominates in the eastern Pacific. That is, changes in w' lead to enhanced Ekman feedback in the eastern Pacific, due to the product $w' \frac{\partial \bar{T}}{\partial z}$ being largest there. We have updated Fig. 2 by placing box B in 130W-80W and JAN-MAR (year +1), where it most accurately describes the contribution of the Ekman feedback to the heat content tendency and El Niño growth as suggested by your comment.

(7) L156-159: Related to the previous comment, it seems to me that the W' -related enhancement of the Ekman feedback during boreal winter (Figure 9f) is more relevant than the mean stratification-related Ekman feedback during boreal spring (Figure 9i) to explain why the 21C El Nino can persist into boreal spring during its decaying phase.

Response: Please note that supplementary Fig. 9 is now supplementary Figs. 6 in the revised manuscript.

We agree. In the original paper, we noted that the contribution from projected changes in the mean stratification in $\frac{\partial \bar{T}}{\partial z}$ (Supplementary Fig. 6) is mostly during the developing and/or decay phase of El Niño, supporting faster growth and slower dissipation of the events.

However, and as you mentioned, the contribution of changes in mean stratification $\frac{\partial \bar{T}}{\partial z}$ is mostly in the developing phase, whereas the changes in the anomalous upwelling w' is more relevant in explaining the persistence of the events into the boreal spring (i.e., decay phase). We have clarified this in the revised manuscript.

(8) L161-162: Related to Comment (7), I cannot agree that “increased positive Ekman feedback between 150°W and 100°W) causes El Niño SSTAs to persist longer in the eastern Pacific in the

late 21C". I think it is the increased positive Ekman feedback between "120W and 80W" during JAN-APR(+1) that should be emphasized to explain the persistent decaying feature. And this increased Ekman feedback can be attributed to the enhanced upwelling anomalies (W') of the 21C El Niño (Figure S9f, rather than Figure S9i). The following analysis should try to explain why the 21C El Niño can induce a stronger W' than the 20C El Niño.

Response: Please refer to our response to comments (6) and (7). We concur that the increased positive Ekman feedback between 120W and 80W during JAN-MAR(+1) readily explains the persistent decaying feature of 21C El Niños relative to 20C events. We have noted these changes in the revised text as well as re-located label B of Fig. 2 to aid in the description of the results. As mentioned earlier, the Ekman feedback changes in this region explains very well the changes in the heat content tendency, leading to persistence SSTA into the boreal spring in the eastern Pacific.

Regarding the question on why the 21C El Niño can induce a stronger W' than the 20C El Niño? We can see that the stronger El Niño events 21C are associated with stronger westerly wind anomalies and anomalous eastward currents (U' , supplementary Fig. 6b), which induce stronger anomalous downwelling through Ekman convergence. This mechanism usually peaks during the peak El Niño phase i.e., DEC-FEB (+1), as seen in supplementary Fig. 6. In addition, it has been previously shown that during El Niño event, anomalous meridional wind drives convergent surface currents that induce an equatorial downwelling anomaly in the equatorial Pacific (Périgaud et al. 1997). The convergence of the anomalous geostrophic mixed-layer currents during El Niño results in anomalous downwelling in the equatorial Pacific (Su et al. 2010).

We added this discussion and updated the revised manuscript following the reviewer's suggestion. Also note that we added more description to figures, including subtitles and updated captions to aid the discussion.

Périgaud, C., Zebiak, S. E., Mélin, F., Boulanger, J. P., & Dewitte, B. (1997). On the role of meridional wind anomalies in a coupled model of ENSO. *Journal of climate*, 10(4), 761-773.

Su, J., Zhang, R., Li, T., Rong, X., Kug, J. S., & Hong, C. C. (2010). Causes of the El Niño and La Niña amplitude asymmetry in the equatorial eastern Pacific. *Journal of Climate*, 23(3), 605-617.

(9) L170-171: I did not find any statements before this part of the manuscript that concluded from Figure 1 that there is a "tendency to have a shift towards stronger eastern Pacific El Niño events in the late 21C". Figures 1d and 1h did suggest that the 21C El Niño has a stronger peak intensity than the 20C El Niño, but the increased SSTAs during Year(0) and the beginning of Year(+1) appear more in the central Pacific rather than in the eastern Pacific. Only during the decaying phase (i.e., Year +1) does the 21C El Niño linger and shift to the eastern Pacific.

Response: We agree that more clarification and motivation need to be provided before the statement in L170-171. As you pointed out, Fig. 1 shows a strengthening of the SSTA during Year (0) in the central Pacific. In addition, there is also a tendency for future El

Niño events to evolve more strongly to the eastern Pacific as shown in Fig. 1c for CMIP6 and Fig. 1d for CESM-LENS when comparing the differences in 21C minus 20C spatio-temporal evolution of SSTA. This is further supported by the eastward shift in convection and precipitation signals shown in Fig. 1g (CMIP6) and Fig. 1h (CESM-LENS).

We also showed that those El Niño events that persist into the boreal spring (i.e., events that are of the eastern Pacific type; Lee et al. 2018, ref # 40) are projected to increase more than any other flavor of El Niño events (Supplementary Fig. 3). The heat budget analysis presented in our work supports the findings that El Niño SSTAs are projected to persist longer in the eastern Pacific in the late 21C, owing to projected changes of major El Niño feedbacks (e.g., increased positive thermocline feedback east of 140°W and negative thermocline feedback west of 140°W, and increased positive Ekman feedback between 120°W and 80°W), and consistent with increase in eastward SSTA propagation during the decay phase of El Niño (Chen et al. 2017, ref #38) and a more effective thermocline feedback in the eastern equatorial Pacific (Carreric et al. 2020, ref#8), as also shown in Supplementary Fig. 8 here.

We have clarified this statement in the revised manuscript by adding a summary of the above reply when describing Fig. 1 in Section 2.

Carréric, A., Dewitte, B., Cai, W., Capotondi, A., Takahashi, K., Yeh, S. W., ... & Guémas, V. (2020). Change in strong Eastern Pacific El Niño events dynamics in the warming climate. *Climate Dynamics*, 54(1-2), 901-918.

Chen, C., Cane, M. A., Wittenberg, A. T., & Chen, D. (2017). ENSO in the CMIP5 simulations: life cycles, diversity, and responses to climate change. *Journal of Climate*, 30(2), 775-801. <https://doi.org/10.1175/JCLI-D-15-0901.1>

Lee, S. K., Lopez, H., Chung, E. S., DiNezio, P., Yeh, S. W., & Wittenberg, A. T. (2018). On the fragile relationship between El Niño and California rainfall. *Geophysical Research Letters*, 45(2), 907-915. <https://doi.org/10.1002/2017GL076197>

(10) L187-188: Similar to the previous comment, are you referring to the decaying phase or the peak phase of the 21C El Niño? Please add the calendar months (e.g., JAN(0), ...) in these statements.

Response: We are referring to the decay phase from FEB to APR(+1) where the thermocline feedback and Ekman feedback produce a positive tendency in the heat content, extending the event into the boreal spring in the Eastern Pacific. We clarified this in the revised text and added more references to the figures by adding specific references to calendar months.

(11) L171-172: Please elaborate what you mean by “The delayed thermocline feedback..”. What is “delayed” referring to here?

Response: We are referring here to how the thermocline feedback plays a dominant role during the decay phase, in the central Pacific (Fig. 2, thermocline panel). Note that there is a strong discharge of heat content (Fig. 2, tendency panel), leading to rapid shoaling of the thermocline and induces a subsurface cooling from the climatological upwelling, which

leads to a strong cooling tendency in SST. This mechanism is explained in Kug et al. 2010 (Ref. 45 of the revised text). However, we agree that the term “delayed” is confusing as thus we removed from the revised manuscript.

Kug, J. S., Choi, J., An, S. I., Jin, F. F., & Wittenberg, A. T. (2010). Warm pool and cold tongue El Niño events as simulated by the GFDL 2.1 coupled GCM. *Journal of Climate*, 23(5), 1226-1239. <https://doi.org/10.1175/2009JCLI3293.1>

(12) Figures S9 and S10: Please consider adding subtitles (or the term from Table S4) to each panel to make it easier to understand what contribution each panel represents. I have to admit that I have a difficult time to completely follow the discussions in L182-197. One main reason is because those discussions were based on the results presented in Figures S9 and S10, each of which include many panels that do not have subtitles. I have to go back and forth between these figures (and Figures 1 and 2) and figure captions (on the other pages of the manuscript) to remind myself what each panel means.

Response: We agree that Figs. S9 and S10 need more labeling to aid with the discussion. We have added more subtitles and updated the figure captions in the revised manuscript. Please note that Fig. S9 is now supplementary Fig. 6 and Fig. S10 is supplementary Fig. 7 in the revised text.

(13) Please consider using a few sentences to summarize the key findings from the heat budget analysis in Section 3.a.

Response: We added the following few sentences to summarize the key findings from the heat budget analysis.

In summary, the projected changes of major El Niño feedbacks (e.g., increased positive thermocline feedback east of 140°W and negative thermocline feedback west of 140°W, and increased positive Ekman feedback between 120°W and 80°W) cause El Niño SSTAs to persist longer in the eastern Pacific in the late 21C, in contrast to a westward propagation of SSTAs typically observed during the late 20C (Figs. 1b and f).

(14) L291-292: What is the basis for the statement that the 21C El Niño “leading to a northeastward intensification of the circulation anomaly”?

Response: This statement is based on the findings of Meehl et al. (2007); Yeh. et. al. (2018); and others which found that projections for the El Niño teleconnection patterns over the U.S. are projected to shift eastward and northward in a warmer climate. This northeastward shift in teleconnection is driven by the expected eastward shift of the PNA teleconnection pattern under global warming (Zhou et al., 2014), resulting from a systematic eastward migration of convection centers associated with both El Niño and La Niña in the future (Power et al., 2013).

With that said, we agree that the clause was not well written and did not specify that this northeastward shift in teleconnection is specific to those affecting North America. We have clarified this in the revised text and included a few relevant references. Also note that we

rewrote section 4 on remote impacts to more succinctly describe ENSO impact changes for North, South America, and Australia separately.

Meehl, G. A., Tebaldi, C., Teng, H., & Peterson, T. C. (2007). Current and future US weather extremes and El Niño. *Geophysical Research Letters*, 34(20).

Power, S., Delage, F., Chung, C., Kociuba, G., & Keay, K. (2013). Robust twenty-first-century projections of El Niño and related precipitation variability. *Nature*, 502(7472), 541-545.

Yeh, S. W., Cai, W., Min, S. K., McPhaden, M. J., Dommenges, D., Dewitte, B., ... & Kug, J. S. (2018). ENSO atmospheric teleconnections and their response to greenhouse gas forcing. *Reviews of Geophysics*, 56(1), 185-206.

Zhou, Z. Q., Xie, S. P., Zheng, X. T., Liu, Q., & Wang, H. (2014). Global warming-induced changes in El Niño teleconnections over the North Pacific and North America. *Journal of Climate*, 27(24), 9050-9064.

(15) L296: Figure 4 reveals that the teleconnections are different between the 20C and 21C El Ninos, but I don't see the reason why we can conclude that there is a "... projected enhancement of the 20C teleconnection into the 21C". The spatial patterns of "21C minus 20C" do not resemble the spatial patterns of the 20C impacts.

Response: Agree. We have rewritten most of section 4 on remote teleconnections in order to deliver a more consistent story. Note that we have separated the remote impacts on North America, South America, and Australia into independent sub-sections. While there is a projected enhancement of the 20C teleconnection into the 21C for some regions, as you noted, the spatial patterns of "21C minus 20C" do not resemble the spatial patterns of the 20C impacts. We have clarified this in the revised text and avoided statements generalizing projected changes. We also added a figure (Supplementary Fig. 10) to depict the changes in remote teleconnections from a circulation perspective by showing geopotential high anomalies and velocity potential composites.

(16) The remote impacts discussed in Section 4 are interesting but lack insights into the physical processes that can link the different evolution patterns of the 21C and 20C El Ninos to their climate impacts over North and South America and Australia. The value of this section would be greatly increased if such insights were provided.

Response: Agree, we have now added more insight details on teleconnection changes associated with future El Niño events. Note that, as suggested by you and reviewer 1, we have added more succinct description of the remote impacts and literature review on previous work. As such we added discussion on previous findings and condensed the results to just two figures (Figs. 4 and 5) by showing results from CESM-LENS and moving the CMIP6 results to the supplementary material. We also included more analysis on changes in circulation anomalies associated with El Niño by looking at composites of upper-level circulation changes (Supplementary Fig. 10).

Reviewer #3 (Remarks to the Author):

Comments for “Projections of Faster Onset and Slower Decay of El Niño in the 21st Century” by Lopez et al.

Summary: This study investigates the change in seasonal evolution of El Niño under global warming. Based on simulations by CESM and CMIP6 models, the authors report that El Niño is projected to develop more quickly in developing stage and persist longer under global warming. Furthermore, the change is considered to affect weather patterns around the globe.

Overall opinion: I think the results are interesting, but I think there are some problems for the paper. First, this study lacks a very clear and deep explanation or physical mechanism for the change. Second, this study lacks enough analyses about the uncertainty of the results, which reduce the confidence of conclusion. The specific comments for the two points are listed as follows.

Response: We appreciate your honest review and hope that the revised manuscript addresses your comments/suggestions. As such, we added more in-depth analysis on the spatio-temporal evolution of El Niño events, underlying mechanisms, and remote effects. For example, we now include a more detailed heat budget analysis, depicting the contribution from non-linear advection terms to the ENSO growth and decay. We also added more supporting references to the remote impacts of El Niño and their future projected changes. Finally, we added more robust statistical significance testing as you suggest in your comments below.

Please find our detailed response to your comments/suggestions below in bold face.

Major comments

1. Although the authors do the mixed layer heat budget and think the change of Ekman feedback and thermocline feedback may be responsible for the change of El Niño seasonal evolution. But the reasons for the changes of some its components, for example w^{\wedge} , $\partial T^{\wedge}/\partial Z$, WWBs, are not clearly given.

Response: We agree that a more succinct explanation on how these changes in the individual components if the feedback should be given. As replied to reviewer # 2, and in the revised text, the stronger El Niño events in 21C are associated with stronger westerly wind anomalies and anomalous eastward currents (u^{\wedge} , supplementary Fig. 5b), which induces stronger anomalous downwelling through Ekman convergence. This mechanism usually peaks during the peak El Niño phase i.e., DEC-FEB (+1), as seen in supplementary Fig. 5. In addition, it has been previously shown that during El Niño event, anomalous meridional wind drives convergent surface currents that induce an equatorial downwelling anomaly in the equatorial Pacific (Périgaud et al. 1997). The convergence of the anomalous

geostrophic mixed-layer currents during El Niño results in anomalous downwelling in the equatorial Pacific (Su et al. 2010).

We have also incorporated further analysis/discussion regarding the mixed layer heat budget analysis to explicitly address the role of the nonlinear ENSO processes in the differences in spatio-temporal El Niño evolution. These non-linear terms were lumped into a residual term in the original version of the paper, but now we are explicitly discussing the role of non-linear advection terms.

The nonlinear advection terms for the heat budget analysis are shown as supplementary Fig. 4 of the revised manuscript (or as Fig. R1 in the reply to Reviewer #1). The residual term still comprises of diffusive heating, unresolved sub-grid scale heat fluxes, and sub-monthly scale heating. Overall, the contribution of the nonlinear terms to the heat budget analysis is much smaller in amplitude than those of the main ENSO feedbacks (e.g., thermocline, zonal advective, and Ekman feedback terms, Fig. 2). However, a projected increase in the nonlinear meridional advection may support the growth of future El Niño events (supplementary Fig. 4, label A) whereas an increase in the nonlinear zonal advection might aid in the persistence of future El Niño events into the boreal spring in the eastern Pacific (supplementary Fig. 4, label B). It should also be noted that the total residual term serves as a damping term of the heat content anomaly (Fig. 2) and its damping role is projected to increase further in the 21C.

Please refer to the revised paper where we added more discussion on changes of each component of the mixed layer heat budget. We also clarified the plots by adding more subtitles to the figures.

Périgaud, C., Zebiak, S. E., Mélin, F., Boulanger, J. P., & Dewitte, B. (1997). On the role of meridional wind anomalies in a coupled model of ENSO. *Journal of climate*, 10(4), 761-773.

Su, J., Zhang, R., Li, T., Rong, X., Kug, J. S., & Hong, C. C. (2010). Causes of the El Niño and La Niña amplitude asymmetry in the equatorial eastern Pacific. *Journal of Climate*, 23(3), 605-617.

In addition, I see a stronger cooling from Jan(1) to Apr(1) in the warming scenario from Fig. 2, which seems contrary to the authors' conclusion of slower dissipation of El Niño under global warming.

Response: Please note that this stronger cooling is in the central and western Pacific rather than the slower dissipation of eastern Pacific events. Note that we mention in the revised paper that the warm El Niño SSTAs in the central Pacific dissipate more rapidly during boreal winter and spring. In contrast, warm El Niño SSTAs in the far eastern Pacific persist throughout boreal spring in 21C, extending the El Niño there in CESM-LENS and inCMIP6. These projected changes mark a contrast in El Niño SSTA evolution and a projected enhancement and eastward shift in convection over the tropical Pacific (Fig. 1). The heat budget analysis presented in our work supports the findings that El Niño SSTAs are projected to persist longer in the eastern Pacific in the late 21C, owing to projected

changes of major El Niño feedbacks (e.g., increased positive thermocline feedback east of 140°W and negative thermocline feedback west of 140°W, and increased positive Ekman feedback between 120°W and 80°W).

In addition, the difference in SSTA evolution from 20C to 21C suggests an increased tendency for eastward propagation of the SSTA under ACC, consistent with previous work, owing to a weakening of the Walker Circulation and associated eastward shift in convection (Santoso et al. 2013).

Please refer to the section 2 of the revised manuscript where this is clarified.

Previous studies show that the interaction between eastern Pacific ITCZ and El Niño in boreal spring is key to determine El Niño decaying speed. Does the interaction contribute to the slow decay of El Niño under global warming?

Response: Agree, a recent study showed that the interaction between the ITCZ and El Niño in boreal spring is a key element in determining the decay of El Niño (Xie et al. 2018). In this season, the eastern Pacific SST reaches its annual maximum (minimum) in the Southern (Northern) Hemisphere. The ITCZ is climatologically further south in this season with two rainfall maxima striding the equator from February-April (Fig. R6 below). This period coincides with the period of slower decaying El Niño events in the eastern Pacific shown in Fig. 1. Note also that during the boreal spring, atmospheric convection in the eastern Pacific is sensitive to small changes in surface temperature as the SST is near the convection threshold. Future projections from CESM-LENS show an enhanced climatological precipitation throughout the ITCZ, including westerly wind anomaly (21C minus 20C) in February-April (Fig. R1c).

Strong El Niño events can cause deep convection in the eastern Pacific (Zheng et al. 2016). This convection produces westerly wind anomalies, allowing for further prolonged and further eastward intrusion of SST anomalies, weakening the mean upwelling and thus extending the persistence of strong El Niño events into the boreal spring (Peng et al. 2020). We can see that the stronger El Niño events in 21C are associated with stronger westerly wind anomalies and anomalous eastward currents (u' , supplementary Fig. 5b), which induces stronger anomalous downwelling through Ekman convergence.

Therefore, the reduction in the equatorial Trade winds in the 21C relative to 20C (i.e., westerly anomalies) suppresses the equatorial upwelling. This coupled to the eastern Pacific warming and enhanced stratification slowdown the decay of El Niño events in the 21C over the eastern Pacific.

We have added this discussion to the revised manuscript (section 3). Fig. R1 below is added as Supplementary Fig. 9.

Peng, Q., Xie, S. P., Wang, D., Kamae, Y., Zhang, H., Hu, S., ... & Wang, W. (2020). Eastern Pacific wind effect on the evolution of El Niño: Implications for ENSO diversity. *Journal of Climate*, 33(8), 3197-3212.

Xie, S. P., Peng, Q., Kamae, Y., Zheng, X. T., Tokinaga, H., & Wang, D. (2018). Eastern Pacific ITCZ dipole and ENSO diversity. *Journal of Climate*, 31(11), 4449-4462.

Figure R6. Latitude-time precipitation (color, mm day^{-1}) and wind stress (vector, N m^{-2}) climatology from CESM-LENS zonally averaged from 140°W - 80°W for the a) 20C period (1951-2000), b) 21C period (2051-2100), and c) the difference between the 21C minus 20C periods. Precipitation and wind stress vectors differences in panel c) that are significant at a 95% confidence level using a bootstrapping subsampling technique are shown by shading color and thick vectors respectively.

2. In this study, the authors use Student's t-test to show the confidence level of their results. Such analyses may be not enough to show that their results are confident. I recommend the author do some further analyses about the uncertainty of the results among different models of members, and give reasons for the uncertainty.

Response: We agree with the reviewer that a more robust method to assign confidence levels should be used. We can take advantage of the multi-ensemble simulations of CESM-

LENS (30 ensembles) and CMIP6 (16-models) to perform a bootstrapping technique to assign confidence levels. This technique, as opposed to the student-T test, is a non-parametric test with the advantage that it does not make assumptions on the distribution of the data being assessed (variance, assumption of normality). As suggested by the reviewer, and in the revised version, we now rely on a bootstrapping method to draw confidence intervals by subsampling the dataset. For example, all composite analyses presented in the revised paper are obtained by randomly selecting $r=15$ out of $n=30$ ensembles from CESM-LENS with replacement (see combination with replacement equation R2 below). This is done 1000 times in order to build a significant distribution of composites and assign 95th percentile confidence levels. Similarly, we follow the same approach when analyzing composites from CMIP6 by selecting $r=8$ out of $n=16$ model ensembles. We have added this analysis in the revised text and a brief description of this method in the Method section of the paper.

$$\binom{n}{r} = \frac{n!}{r!(n-r)!} = \textit{possible combinations} \quad (R2)$$

REVIEWERS' COMMENTS

Reviewer #1 (Remarks to the Author):

The authors have addressed my previous comments. I have only a few minor points below.

It is helpful that the nonlinear terms of the heat budget are now discussed. However, note that nonlinear terms are associated with ENSO extremes (related to asymmetry, diversity), and so expectedly they would appear small when averaged across all ENSO events. This needs to be clarified to avoid confusion, e.g., add the following phrases within the arsterisks:

“Nonlinear processes have been shown to be important for understanding the ENSO response to mean state changes from anthropogenic forcing⁹, **as they govern characteristics of extreme ENSO events^{36,41}**, ...” <- also add these references (Cai et al. 2022, Santoso et al. 2013).

“Overall, the contribution of the nonlinear terms to the heat budget analysis is much smaller in amplitude than the linear ENSO feedback terms (e.g., thermocline, zonal advective, and Ekman feedback terms, Fig. 2) **when averaged across all El Niño events in general**”.

L28: “Previous studies based on the climate models participating in the Coupled Model Intercomparison Project Phase 5 (CMIP5) have suggested an increase in the frequency of extreme El Niño events in the 21st Century (21C) in response to increasing greenhouse gases^{5,6,7,8}, as well as an increase in ENSO amplitude⁹, and associated rainfall variability^{10,11,12}” – Please clarify up front that many of these findings are also found in CMIP6 (Cai et al. 2021).

L451: “All models employed here show relatively accurate representation of El Niño...” - I found this paragraph could be written better. It is known that it is not the case, e.g., ENSO asymmetry is underestimated in climate models, including CMIP6 (Cai et al., McKenna et al. 2020 10.1038/s41598-020-68268-9). So it is better to be specific, e.g.,: “All models employed here show reasonable representation of spatio-temporal evolution of El Niño events (Supplementary Fig. 13), although differences among individual models are notable (Supp. Fig. 14)”.

L349: “wave trans” should be “wave trains”

Please check the referencing throughout. E.g., in L326 Ref. 40 should be Ref. 41.

Reviewer #2 (Remarks to the Author):

The authors have satisfactorily addressed my comments. I think the revised manuscript can be accepted for publication.

Reviewer #3 (Remarks to the Author):

I am glad to review this paper again. The authors have addressed most my previous question. The paper is improved in this round.

I am especially interested in the result that the El Niño likely persist longer under global warming, as so many climate anomalies are related to the persistence of El Niño. In the revision, the authors suggest that the increasing persistence of El Niño under global warming is due to an enhanced Ekman feedback in the eastern Pacific, which is in turn mainly due to an enhanced downwelling associated with El Niño (W'). On one hand, stronger Ekman feedback can make El Niño persist longer via reducing damping. On the other hand, a longer-persisted El Niño can maintain equatorial westerly wind and lead to a stronger Ekman feedback. From this point, they are just feedback process, but couldn't determine whether El Niño persist longer or shorter. So, what is the determining factor to control El Niño's persistence under global warming? I would like the author could address it.

REVIEWERS' COMMENTS

Please extend our gratitude to the three anonymous reviewers for their constructive comments/suggestions that have led to a significant improvement to our manuscript.

In what follows, our responses are highlighted in boldface, while the original reviewers' suggestions are in normal font.

Reviewer #1 (Remarks to the Author):

The authors have addressed my previous comments. I have only a few minor points below.

Response: We are very grateful for your constructive comments/suggestions and hope to have successfully addressed all your inquiries.

It is helpful that the nonlinear terms of the heat budget are now discussed. However, note that nonlinear terms are associated with ENSO extremes (related to asymmetry, diversity), and so expectedly they would appear small when averaged across all ENSO events. This needs to be clarified to avoid confusion, e.g., add the following phrases within the arterisks:

“Nonlinear processes have been shown to be important for understanding the ENSO response to mean state changes from anthropogenic forcing⁹, ****as they govern characteristics of extreme ENSO events^{36,41**}, ...**” <- also add these references (Cai et al. 2022, Santoso et al. 2013).

“Overall, the contribution of the nonlinear terms to the heat budget analysis is much smaller in amplitude than the linear ENSO feedback terms (e.g., thermocline, zonal advective, and Ekman feedback terms, Fig. 2) ****when averaged across all El Nino events in general****”.

Response: We added the suggested text and reference to the revised manuscript.

L28: “Previous studies based on the climate models participating in the Coupled Model Intercomparison Project Phase 5 (CMIP5) have suggested an increase in the frequency of extreme El Niño events in the 21st Century (21C) in response to increasing greenhouse gases^{5,6,7,8}, as well as an increase in ENSO amplitude⁹, and associated rainfall variability^{10,11,12}” – Please clarify up front that many of these findings are also found in CMIP6 (Cai et al. 2021).

Response: We added the suggested text and reference to the revised manuscript.

L451: “All models employed here show relatively accurate representation of El Nino...” - I found this paragraph could be written better. It is known that it is not the case, e.g., ENSO asymmetry is underestimated in climate models, including CMIP6 (Cai et al., McKenna et al. 2020 10.1038/s41598-020-68268-9). So it is better to be specific, e.g.,: “All models employed here show reasonable representation of spatio-temporal evolution of El Nino events (Supplementary Fig. 13), although differences among individual models are notable (Supp. Fig. 14)”.

Response: We added the suggested text to the revised manuscript.

L349: “wave trans” should be “wave trains”

Response: Fixed

Please check the referencing throughout. E.g., in L326 Ref. 40 should be Ref. 41.

Response: Fixed. Please note that Ref. 41 is now moved up to Ref. 13 (Cai et al. 2021).

Reviewer #2 (Remarks to the Author):

The authors have satisfactorily addressed my comments. I think the revised manuscript can be accepted for publication.

Response: We are very grateful for your constructive comments/suggestions.

Reviewer #3 (Remarks to the Author):

I am glad to review this paper again. The authors have addressed most my previous question. The paper is improved in this round.

Response: We are very grateful for your constructive comments/suggestions and hope to have successfully addressed all your inquiries.

I am especially interested in the result that the El Nino likely persist longer under global warming, as so many climate anomalies are related to the persistence of El Nino. In the revision, the authors suggest that the increasing persistence of El Nino under global warming is due to an enhanced Ekman feedback in the eastern Pacific, which is in turn mainly due to an enhanced downwelling associated with El Nino (W'). On one hand, stronger Ekman feedback can make El Nino persist longer via reducing damping. On the other hand, a longer-persisted El Nino can maintain equatorial westerly wind and lead to a stronger Ekman feedback. From this point, they are just feedback process, but couldn't determine whether El Nino persist longer or shorter. So, what is the determining factor to control El Nino's persistence under global warming? I would like the author could address it.

Response: The persistence of future El Niño events into the boreal spring is addressed via heat budget analysis presented in the revised version, which included all source terms determining the temperature tendency and thus the evolution of the spatio-temporal mixed-layer temperature anomaly associated with El Niño. Below is a brief description of our findings regarding the determining factor to control El Niño's persistence under global warming as presented in the revised text:

The Ekman feedback is enhanced in the eastern equatorial Pacific between 120°W and 80°W (point labeled B in Fig. 2) supporting the persistence of future El Niño events into the boreal spring in the eastern Pacific. As noted in section 3 of the revised text, the projected increase of the Ekman feedback can be explained by both contributions from anomalous upwelling (w' , Supplementary Fig. 5 and Fig. 6f) and an increase in the mean stratification ($\frac{\partial \bar{T}}{\partial z}$, Supplementary Fig. 5 and Fig. 6i). In addition, the mean thermocline is projected to deepen in the eastern Pacific and sharpen (e.g., increased $\frac{\partial \bar{T}}{\partial z}$) during the 21C (Supplementary Fig. 8), enhancing the thermocline and Ekman feedbacks and thus amplifying El Niño events towards the eastern Pacific during the decay phase (Fig. 1c and d). We also found an increase in the nonlinear zonal advection (point labeled B in Supplementary Fig. 4), driven by anomalous eastward currents (u' , Supplementary Fig. 5b), and stronger westerly wind burst anomalies (Fig. 3h), which drives the persistence of the events into the boreal spring. These feedbacks are a central component determining the temporal evolution of temperature in the mixed layer and thus in the heat content growth and decay during the life cycle of an El Niño event.

However, and as you suggested, there needs to be further explanation of the determining factor that control El Niño's persistence into the boreal spring. In this season, as discussed in section 3b ("Impacts of a changing tropical Pacific climatology") in the eastern Pacific SST reaches its annual maximum in the Southern Hemisphere; thus, the ITCZ is climatologically further south in boreal spring (Supplementary Fig. 9). We found that deep convection associated with El Niño events in the 21C is greatly enhanced and produces stronger westerly wind anomalies, allowing for more prolonged and eastward-intruding SST anomalies, weakening the mean upwelling and thus extending the persistence of El Niño events into boreal spring. This is consistent with previous work, which found that strong eastern Pacific events that persist into the boreal spring are strongly coupled to the ITCZ (refs. # 8 and 48: Carréric et al. 2020; Peng et al. 2020) and that the interaction between the ITCZ and El Niño in boreal spring is a key element in determining the decay of El Niño, as discussed in ref # 47 (Xie et al. 2018). That is, the reduction in the equatorial Trade winds in the 21C relative to 20C (i.e., westerly anomalies) suppresses the equatorial upwelling. This, together with the eastern Pacific warming and enhanced stratification, slow down the decay of El Niño events in the 21C in the eastern Pacific, consistent with (ref. # 48: Peng et al. 2020).